# The genetic architecture of sporadic and multiple consecutive miscarriage

Triin Laisk 🆔 et al.[#]

Miscarriage is a common, complex trait affecting ~15% of clinically confirmed pregnancies. Here we present the results of large-scale genetic association analyses with 69,054 cases from five different ancestries for sporadic miscarriage, 750 cases of European ancestry for multiple (≥3) consecutive miscarriage, and up to 359,469 female controls. We identify one genome-wide significant association (rs146350366, minor allele frequency (MAF) 1.2%, $P = 3.2 \times 10^{-8}$, odds ratio (OR) = 1.4) for sporadic miscarriage in our European ancestry meta-analysis and three genome-wide significant associations for multiple consecutive miscarriage (rs7859844, MAF = 6.4%, $P = 1.3 \times 10^{-8}$, OR = 1.7; rs143445068, MAF = 0.8%, $P = 5.2 \times 10^{-9}$, OR = 3.4; rs183453668, MAF = 0.5%, $P = 2.8 \times 10^{-8}$, OR = 3.8). We further investigate the genetic architecture of miscarriage with biobank-scale Mendelian randomization, heritability, and genetic correlation analyses. Our results show that miscarriage etiopathogenesis is partly driven by genetic variation potentially related to placental biology, and illustrate the utility of large-scale biobank data for understanding this pregnancy complication.

---

[#]A list of authors and their affiliations appears at the end of the paper.

Miscarriage is defined by the World Health Organization (WHO) as the spontaneous loss of an embryo or fetus weighing <500 g, up to 20–22 weeks of gestation[1]. Recurrent miscarriage, which is considered to be a more severe phenotype, is currently defined as two or more miscarriages[1], although previous definitions also include three or more (consecutive) miscarriages[2,3]. It is acknowledged that using two miscarriage as the definition of recurrent miscarriage is at least in part to facilitate research, improve shared decision making with patients and provide them psychological support, rather than because of more specific evidence that this defined a unique phenotype[1], leaving open the question where to draw the line for separating different miscarriage phenotypes from a biological point of view.

Miscarriage is the most common complication of pregnancy[4,5] and the majority of miscarriages, both sporadic or recurrent[6,7], happen in the first trimester[7,8]. Miscarriage is associated with excessive bleeding, infection, anxiety, depression[9], infertility[10], and an increased lifetime risk of cardiovascular disease[11,12].

The risk of miscarriage increases with maternal age[4], and has been associated with a range of causes; embryo and oocyte aneuploidy, parental chromosomal abnormalities, maternal thrombophilias, obesity, and endocrine and immunological dysregulation[6] but causal underlying factors remain largely unknown. Miscarriage has a genetic component[13,14], with most studies focusing on associations of maternal genetic variants with recurrent miscarriage. A recent systematic review illustrates the small sample sizes of these studies (vast majority <200 cases) and the heterogeneous definition of cases, and as a consequence identified largely inconsistent results[15].

To discover and map the maternal genetic susceptibility and underlying biology of miscarriage, we combined genome-wide association study (GWAS) results of up to 69,054 cases from different ancestries (European, Chinese, UK South-Asian, UK African, African American, Hispanic American, UK Caribbean) for sporadic miscarriage, and subsequently the results of 750 cases of European ancestry for more severe multiple consecutive miscarriage in the largest genetic study of miscarriage to date. Although there is a continuous discussion on where to draw the line for classifying miscarriages as recurrent, we aimed to capture the more severe end of the phenotypic distribution and to differentiate severe cases from sporadic miscarriage, and potentially identify any differences in the underlying genetic architecture for these two conditions[3,7,16,17], and thus defined sporadic miscarriage as 1–2 miscarriages and multiple consecutive miscarriage as having had ≥3 self-reported consecutive miscarriages[7,18], or the International Classification of Diseases (ICD-10) diagnosis code N96 for habitual abortion (Supplementary Note 1).

We identify one genome-wide significant association for sporadic miscarriage and three genome-wide significant associations for multiple consecutive miscarriage. We further investigate the genetic architecture of miscarriage with biobank-scale Mendelian randomization (MR), heritability, and genetic correlation analyses. Our results show that miscarriage etiopathogenesis is partly driven by genetic variation potentially related to placental biology and illustrate the utility of large-scale biobank data for understanding this pregnancy complication.

## Results

**GWAS meta-analysis**. We first performed a trans-ethnic GWAS meta-analysis for sporadic miscarriage, including genotype data for 69,054 cases and 359,469 female controls (Fig. 1, Supplementary Data 1 and 2). Association summary statistics were aggregated using trans-ethnic meta-regression implemented in the MR-MEGA software[19] for GWAS meta-analysis. After post

GWAS filtering for variants present in at least half ($n = 11$) of the 21 datasets, the trans-ethnic GWAS meta-analysis of 8,664,066 variants revealed a genome-wide significant locus on chromosome 7 (lead signal rs10270417, MAF = 1.7%, $P_{meta} = 6.0 \times 10^{-9}$; Supplementary Data 3, Supplementary Fig. 2), driven by the Kadoorie Chinese-ancestry cohort ($OR_{EUR} = 1.0$ (0.9–1.0); $OR_{Kadoorie} = 86.1$ (21.1–350.3)). However, since it is known that the software used for cohort-level association testing in the China Kadoorie biobank (BOLT-LMM) can overestimate significance for rare SNPs (MAF < 1%) if the case fraction is <10%[20] ($MAF_{Kadoorie} = 0.04\%$, case fraction 8.9%), and the variant was absent from other Chinese-ancestry cohorts (BGI and $UKBB_{CHI}$) due to low MAF, the variant was not taken forward for further analysis and interpretation. A population-specific effect cannot be ruled out but would require local replication.

We also performed a European ancestry only meta-analysis using METAL[21], in 49,996 sporadic miscarriage cases and 174,109 female controls. After filtering for variants present in more than half of the 13 European ancestry cohorts (9,088,459 SNPs), we detected one genome-wide significant locus on chromosome 13 (rs146350366, MAF = 1.2%, $P_{meta} = 3.2 \times 10^{-8}$, OR = 1.4 (1.2–1.6); Fig. 2, Supplementary Data 3, Supplementary Fig. 3).

Next, we performed a European ancestry only meta-analysis aggregating summary statistics in 750 multiple consecutive miscarriage cases and 150,215 controls from three participating cohorts (UKBB, EGCUT, ALSPAC), using Stouffer's Z-score method implemented in METAL[21], as the effect estimates from different cohorts were not directly comparable. Meta-analysis results were filtered to keep variants with an average MAF ≥ 0.5%, cohort-level MAF ≥ 0.1%, and that were present in at least two cohorts ($n = 8,956,145$). Four of the genome-wide significant signals (on chromosomes 2, 9, 11, and 21) were present in all three cohorts and had the same direction of effect (Fig. 3, Supplementary Data 3). Next, we applied the Firth test for significant variants to control for case-control imbalance, and to obtain uniform cohort-level association statistics and a summary effect estimate. This left us with three genome-wide significant signals: on chromosome 9 (rs7859844, MAF = 6.4%, $P_{meta} = 1.3 \times 10^{-8}$, $P_{Firth} = 2.0 \times 10^{-9}$, OR = 1.7 (1.4–2.0)), chromosome 11 (rs143445068, MAF = 0.8%, $P_{meta} = 5.2 \times 10^{-9}$, $P_{Firth} = 1.8 \times 10^{-10}$, OR = 3.4 (2.4–5.0)), and 21 (rs183453668, MAF = 0.5%, $P_{meta} = 2.8 \times 10^{-8}$, $P_{Firth} = 2.5 \times 10^{-9}$, OR = 3.8 (2.4–5.9)). The signal on chromosome 2 (rs138993181, MAF = 0.6%, $P_{meta} = 1.6 \times 10^{-8}$), did not remain significant after the Firth test ($P_{Firth} = 1.7 \times 10^{-7}$, OR = 3.6 (2.2–5.8)) (Supplementary Fig. 4a–d) and was not taken further for functional annotation analysis.

To clarify the potential genetic overlap between miscarriage phenotypes, we performed a cross-phenotype look-up of the associated loci. All the multiple consecutive miscarriage loci were statistically not significant in the spontaneous miscarriage analysis, and vice versa (see Supplementary Note 1). This indicates the genetic basis of sporadic and multiple consecutive miscarriage does not overlap, at least not for the reported loci, and lends further support to our phenotype definitions.

**No overlap with previously proposed candidate genes**. To our knowledge, no previous GWAS for recurrent or sporadic miscarriage have been carried out, but we checked the results for the 333 variants from a previous meta-analysis of published idiopathic recurrent miscarriage candidate gene associations[15] in our European ancestry meta-analyses for both sporadic and multiple consecutive miscarriage. None of these variants were genome-wide significant in either the sporadic or multiple consecutive miscarriage analysis, and only 14 (4.2%) and 11 (3%) were

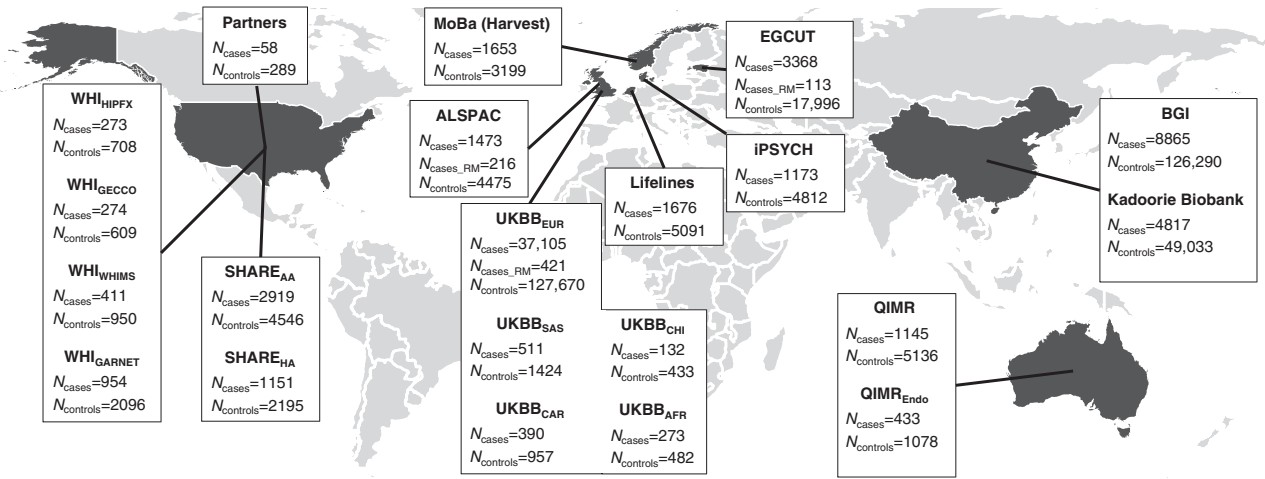

**Fig. 1 Overview of the included cohorts.** Our trans-ethnic GWAS meta-analysis for sporadic miscarriage included data for 69,054 cases and 359,469 female controls, whereas our European ancestry only analysis included 49,996 sporadic miscarriage cases and 174,109 female controls. We also analyzed data for 750 multiple consecutive miscarriage cases, all of European ancestry.

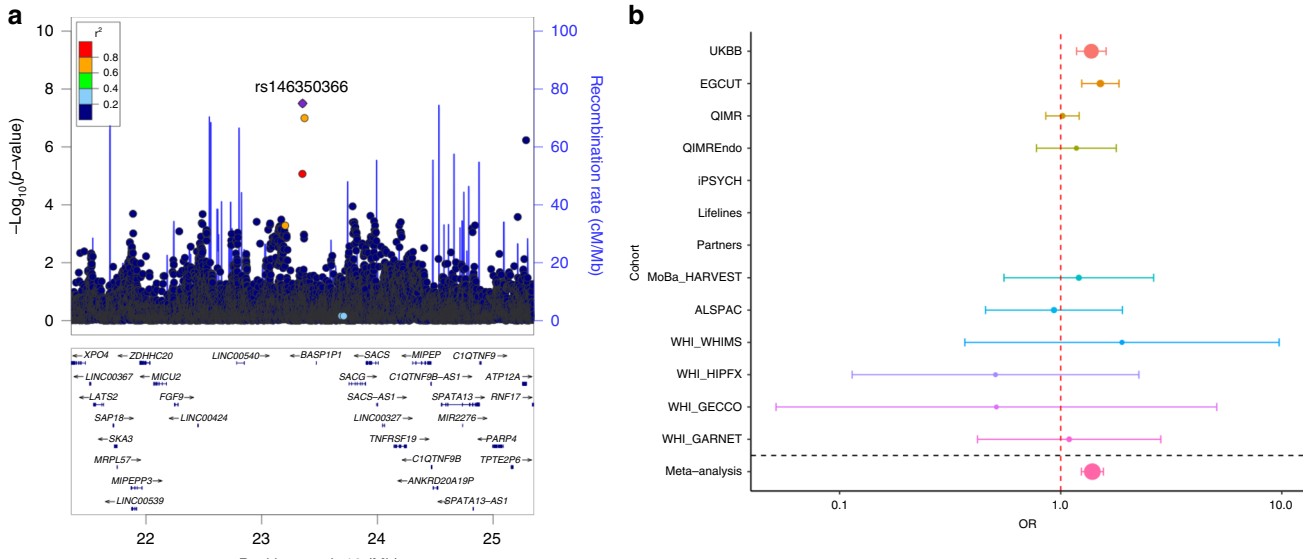

**Fig. 2 Results of the sporadic miscarriage European ancestry GWAS meta-analysis. a** We detected one genome-wide significant locus on chromosome 13 (lead signal rs146350366, $n_{cases} = 49,996$, $n_{controls} = 174,109$). The 1000G EUR reference was used for plotting. **b** The forest plots depict effect estimates in individual cohorts and the size of the dot is proportional to effective sample size (calculated as 4/((1/N_cases)+(1/N_controls))). Error bars depict 95% confidence intervals. Dashed red line indicates the line of no effect. Details of association testing in individual cohorts can be found in Supplementary Data 2. Summary effect estimate from inverse variance fixed effects meta-analysis.

nominally significant ($P < 0.05$) in the respective analyses (Supplementary Data 4), which is less than expected by chance (5%). Two genome-wide linkage scans, one of 44 recurrent miscarriage cases and 44 controls and the other of 38 sibling pairs affected by idiopathic recurrent miscarriage, reported loci on 6q27, 9q33.1, Xp22.1[22] and 3p12.2, 9p22.1 and 11q13.4[13] as associated with idiopathic recurrent miscarriage but none of the multiple consecutive or sporadic miscarriage associations identified in our much larger analysis overlap with these previously reported associations.

**Heritability of miscarriage.** While previous studies have shown preliminary evidence that (recurrent) miscarriage has a genetic predisposition[13,14,22], the heritability of miscarriage and related phenotypes has remained unquantified. We were unable to estimate the heritability for multiple consecutive miscarriage robustly

due to a relatively small number of cases. However, we estimated the traditional heritability of 'ever having miscarried' under a classical twin model using the QIMR twin dataset, including 1853 monozygotic (MZ) and 1177 dizygotic (DZ) complete twin pairs and 2268 individuals from incomplete pairs, and found a heritability of 29% (95% CI 20–38%) for any miscarriage (Supplementary Data 5). In parallel, we used the sporadic miscarriage European ancestry GWAS meta-analysis summary statistics and the LD Score regression (LDSC) method[23] to calculate the SNP-based heritability for sporadic miscarriage. We found the SNP-based heritability estimate to be small, with $h^2 = 1.5\%$ (SE 0.4) on the liability scale (assuming a population prevalence 20%). Similar to other complex traits, our findings show the SNP-heritability is substantially lower than the traditional heritability, suggesting that other sources of genetic variation may have a larger contribution.

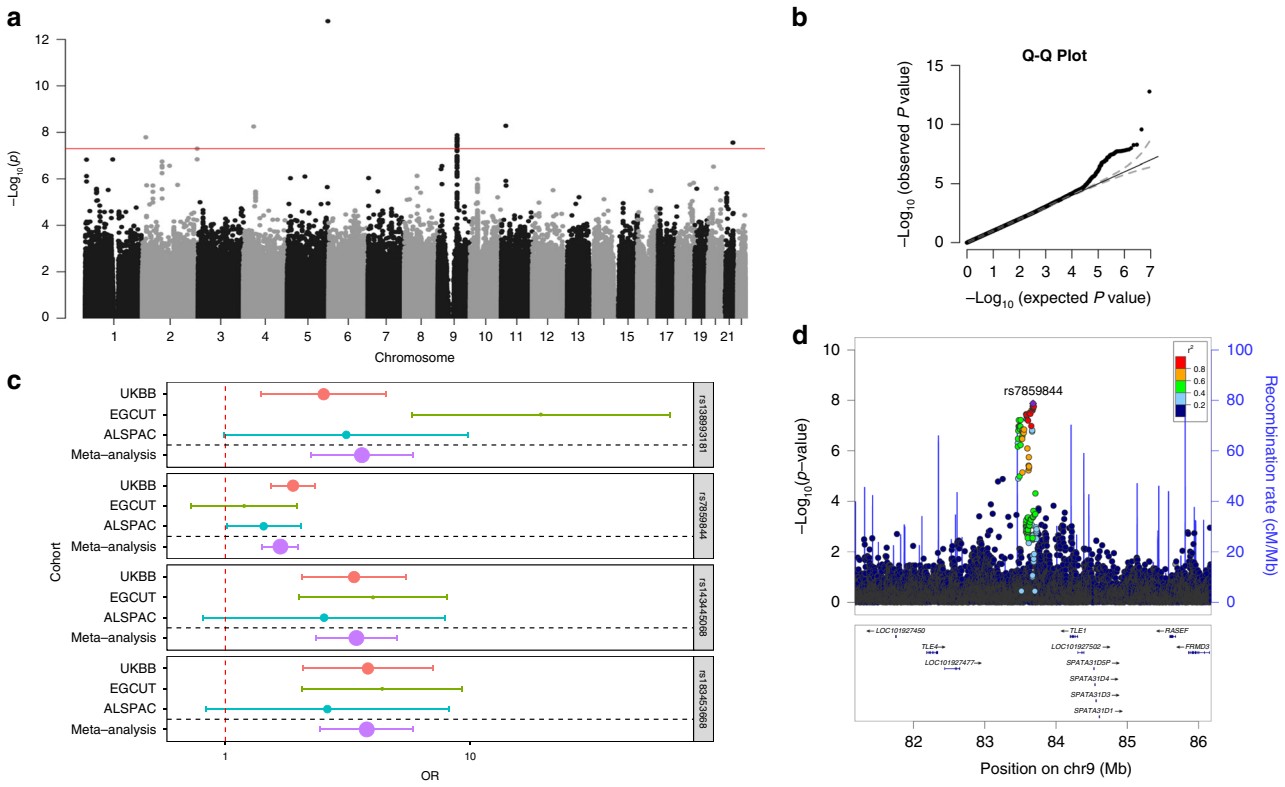

**Fig. 3 Results of the multiple consecutive miscarriage GWAS meta-analysis. a**, **b** Meta-analysis results were filtered to keep variants with an average MAF ≥ 0.5%, cohort-level MAF ≥ 0.1%, and that were present in at least two cohorts ($n = 8,956,146$); **c** Four of the genome-wide significant signals (rs138993181 on chromosome 2, rs7859844 on chromosome 9, rs143445068 on chromosome 11, and rs183453668 on chromosome 21) were present in all three cohorts ($n_{cases} = 750$, $n_{controls} = 150,215$) and had the same direction of effect, effect estimates from Firth test. OR-s (dot) and 95% CI-s are plotted (error bars), the size of the dot is proportional to the effective sample size. **d** Regional plot of the association signal on chromosome 9.

**Miscarriage genetically correlated with number of children**. We assessed pairwise genetic correlations ($r_g$) between sporadic miscarriage and 760 other traits (Supplementary Data 6) available from the LD Hub[24] and found 82 significant (FDR < 0.05) genetic correlations with European-ancestry sporadic miscarriage analysis. For example, significant genetic correlations were observed with reproductive traits (number of children ($r_g = 0.69$, se = 0.12, FDR = $2.0 \times 10^{-6}$) and age at first birth ($r_g = -0.40$, se = 0.10, FDR = $1.1 \times 10^{-3}$)) (Supplementary Data 6). The positive genetic correlation between sporadic miscarriage and number of children is consistent with observational associations between sporadic miscarriage and greater number of live births[25], and with our own observations from phenotype exploration analyses in the UKBB (Supplementary Note 1, Supplementary Fig. 1). We also observed significant correlations with traits related to smoking behavior, mental health, and general well-being (Supplementary Data 6).

**Miscarriage is associated with a variety of health outcomes**. We also examined phenotypic associations of sporadic and multiple consecutive miscarriage with ICD-coded disease outcomes from linked hospital episode statistics in the UKBB dataset, adjusting for number of live births and woman's age and using FDR corrected P-values. We focused only on outcomes with at least one observation among the cases, resulting in testing >6000 ICD codes for sporadic and >1000 ICD codes for multiple consecutive miscarriage. For sporadic miscarriage, the majority of associations were related to pregnancy, childbirth and the puerperium (P-values ranging between $9.9 \times 10^{-79}$ and $4.4 \times 10^{-2}$; Supplementary Data 7; Supplementary Fig. 5). Sporadic miscarriage was also positively associated with a wide variety of diagnoses, including

asthma ($P = 1.6 \times 10^{-20}$, OR = 1.2 (1.19–1.3)), stillbirth ($P = 5.1 \times 10^{-5}$, OR = 74.3 (10.0–549.2)), depressive episodes ($P = 1.4 \times 10^{-7}$, OR = 1.2 (1.1–1.3)), irritable bowel syndrome ($P = 3.5 \times 10^{-9}$, OR = 1.3 (1.2–1.4)), intentional self-poisoning ($P = 9.5 \times 10^{-4}$, OR = 1.6 (1.2–2.0)) and negatively associated with endometrial cancer ($P = 9.9 \times 10^{-3}$, OR = 0.8 (0.7–1.0)). Multiple consecutive miscarriage was positively associated with tubulointerstitial nephritis ($P = 7.8 \times 10^{-5}$, OR = 5.3 (2.3–12.0)), infertility ($P = 1.9 \times 10^{-18}$, OR = 7.5 (4.8–11.7)), ectopic pregnancy ($P = 6.7 \times 10^{-17}$, OR = 25.4 (12.1–53.4)), and others (Supplementary Data 8, Supplementary Fig. 5). Due to the definitions used to extract cases with three or more consecutive miscarriages from self-reported data, this group has less children compared to the other groups (Supplementary Note 1, Supplementary Fig. 1, Supplementary Data 1) and thus represents a miscarriage subphenotype with a more severe effect on fertility, which could explain the correlation with infertility diagnosis.

Although it would be interesting to evaluate the observed correlations on a genetic level, sufficiently sized datasets were not available for the majority of these diagnoses on the LD Hub[24] for the UKBB (or published previously). However, we did see some support from our genetic correlation analyses (Supplementary Data 6) for depression ('depressive symptoms', FDR = 0.021, $r_g$ = 0.32, se = 0.1) and asthma (self-reported asthma, FDR = 0.018, $r_g$ = 0.24, se = 0.07).

Previously proposed risk factors for miscarriage include smoking[26], alcohol use[27], and body mass index (BMI)[28]. To explore the possible causal effect of smoking, alcohol, and BMI on sporadic miscarriage, we used a two-sample MR approach[29]. Summary statistics for alcohol use (drinks per week) and smoking initiation (ever smoked regularly) were obtained from the most

recent GWAS for these traits[30] and for BMI were obtained from Locke et al. (2015)[31]. Summary associations between each variant included in the MR analyses and sporadic miscarriage were obtained from the European-ancestry meta-analysis. We harmonized the summary datasets and used inverse variance weighted (IVW)[32], weighted median (WM)[33], and MR-Egger[34] methods to obtain a pooled estimate of the associations of genetic variants for smoking, alcohol use and BMI with SM. Results from IVW showed evidence of a causal association between smoking and SM (OR 1.16, 95% CI 1.11; 1.22), which were consistent with results from WM (OR 1.16, 95% 1.09; 1.24) but not with the point estimate from MR-Egger (OR 0.95, 95% CI 0.79; 1.14), though the 95% confidence intervals overlap. The *P*-value for MR-Egger intercept was 0.029, suggesting that IVW result is likely to be biased by unbalanced horizontal pleiotropy. We did not observe any consistent effect of BMI or alcohol use on SM (Supplementary Data 9; Supplementary Fig. 6). It is important to note that the samples were not independent, and there was some overlap between the risk factor and sporadic miscarriage GWAS samples, especially with the alcohol analyses sample (the percentage of the sporadic miscarriage samples that were in the exposure samples were 10%, 16%, and 38% for BMI, smoking, and alcohol, respectively). This might introduce bias in the results and inflate type I error; MR-Egger estimates are more prone to bias due to sample overlap than IVM and WM[33]. This may explain some of the inconsistency that we see between the MR-Egger and other results for alcohol and possibly those for smoking. Taking these findings together, our analyses suggest that smoking may causally increase the risk of SM, but we cannot exclude the possibility of horizontal pleiotropy explaining some of this effect.

We also conducted a hypothesis generating phenome-wide MR analysis of multiple consecutive miscarriage (using a per allele genetic risk score from the GWAS significant SNPs) in relation to 17,037 diseases and health related traits using the PHESANT[35] package in UKBB ($n = 168,763$) (Supplementary Fig. 7), but identified no robust causal associations (Supplementary Fig. 8, Supplementary Data 10).

**Gene prioritization identifies links with placental biology.** In order to refine the list of candidate genes identified by eQTL and chromatin interaction mapping, we used constraint measures (pLI score[36] and observed/expected (oe) ratio) from gnomAD v2.1.1[37]. The pLI score reflects the probability that a given gene is loss-of-function (LoF) intolerant, with scores ≥0.9 reflecting extreme intolerance to protein-truncating variation, which could point to a reproductive disadvantage in heterozygous LoF cases[36]. The oe ratio is the ratio of the oe number of LoF variants in that gene. It is a continuous measure, where low oe values indicate the gene is under stronger selection for that kind of variation compared to a gene with higher values. The upper bound of the oe 90% CI < 0.35 is recommended for thresholding. The candidate gene list was also compared to information on mouse phenotypes based on two databases[38,39], and against a recently published list of genes potentially relevant to infertility or recurrent intrauterine death[40].

For the sporadic miscarriage European ancestry meta-analysis signal on chromosome 13, a total of five candidate SNPs and 47 potentially causal genes were suggested by chromatin interaction data from 21 different tissues/cell types, while no significant eQTL associations were detected using FUMA[41] (Supplementary Data 11 and 12). Of the protein-coding genes, *FGF9* had the highest pLI score (0.91) and lowest oe ratio (Supplementary Data 15), indicating the gene is LoF intolerant, which is in line with data from mouse models, where homozygous mutants exhibit embryonic and preweaning lethality. Capture Hi-C data

from different cells (including trophoblast-like cells, Supplementary Data 12)[42] showed connections between the GWAS meta-analysis association signal and the *FGF9/MICU9* locus (Fig. 4a). FGF9 is needed for creating a favorable microenvironment for embryo implantation/pregnancy maintenance, possibly either by regulating angiogenesis or participating in recognizing competent blastocysts via the maternal FGF9/embryonic FGFR2 axis[43]. If this 'embryo sensoring' is compromised, implantation of low-quality embryos can occur and potentially result in miscarriage[44]. In addition, FGF9 plays a paracrine role in ovarian progesterone production[45], and has been found to be upregulated at the mRNA level in the endometrium of women with miscarriage[46].

For the multiple consecutive miscarriage association signal on chromosome 9, 53 candidate SNPs and a total of 50 candidate genes were identified by chromatin interaction data: among them protein-coding genes *TLE1*, *TLE4*, *PSAT1*, *IDNK*, *GNAQ*, *RASEF*, *SPATA31D1*, and *FRMD3* (Supplementary Data 13 and 14). Of these, *TLE1*, *TLE4*, and *GNAQ* have pLI scores > 0.9 (Supplementary Data 15). Hi-C data[42] showed interactions between the associated locus and *TLE1* (and between *TLE1* and *TLE4*) (Fig. 4b). Both TLE1 and TLE4 are repressors of the canonical WNT signaling pathway, and participate in controlling placental extravillous trophoblast motility[47]. Invasion of extravillous trophoblast cells into maternal decidua is essential for proper placenta formation, and dysregulation of this process has been associated with several pregnancy complications, including miscarriage[48]. Additionally, there is evidence TLE1/TLE4 may also regulate gonadotropin-releasing hormone (GnRH) expression and/or differentiation of GnRH neurons[49], providing a potential link with gonadotropin regulation. Further, a member of the same family of proteins (Groucho/Transducin-Like Enhancer of Split), TLE6 has been shown to be associated with early embryonic lethality[50] and is known to antagonize TLE1-mediated transcriptional repression[51]. On chromosome 11, both rs143445068 and rs140847838 were highlighted as potential candidate SNPs in the associated region located in the intron of *NAV2*. Chromatin interaction mapping highlighted another 17 candidate genes in the locus, including *DBX1*, *HTATIP2*, *E2F8*, *ZDHHC13*, and *MRGPRX2* (Supplementary Fig. 10). Both *NAV2* and *E2F8* show evidence of constraint, and furthermore, *NAV2* is listed as one of the candidate genes potentially relevant for infertility or recurrent fetal death[40] (Supplementary Data 15). However, there is also strong evidence to support *E2F8* as a potential candidate gene in this locus, as it is required for placental development[52] and fetal viability, and speculated to suppress the invasiveness of extravillous trophoblasts in humans. The E2F transcription factor family has also been proposed as a key regulator of placental genes differentially expressed in recurrent pregnancy loss[53]. Finally, for the association signal on chromosome 21, no other candidate SNPs in addition to the lead signal rs183453668 were identified, and a total of 10 candidate genes were suggested by chromatin interaction data, including *SIK1*, *U2AF1*, *CRYAA*, *HSF2BP*, and *RRP1B* (Supplementary Fig. 11). Of these, *U2AF1* and *SIK1* exhibit intolerance to LoF (Supplementary Data 15), and *SIK1* is associated with early lethality in mouse and has also been shown to play a role in trophoblast differentiation[54].

We then tested for colocalization between multiple consecutive miscarriage and plasma protein levels[55] and expression levels in 49 different tissues[56,57] using coloc[58] (Supplementary Methods). Colocalization can highlight potential mediating genes and tissues by investigating the likelihood of a shared causal variant between a disease and e.g. quantitative trait loci[58]. There was no evidence for colocalization for any of the risk locus–gene/protein–tissue combinations, potentially reflecting that the risk is mediated through other genes than those investigated, or lack of data from the relevant tissue or time-point.

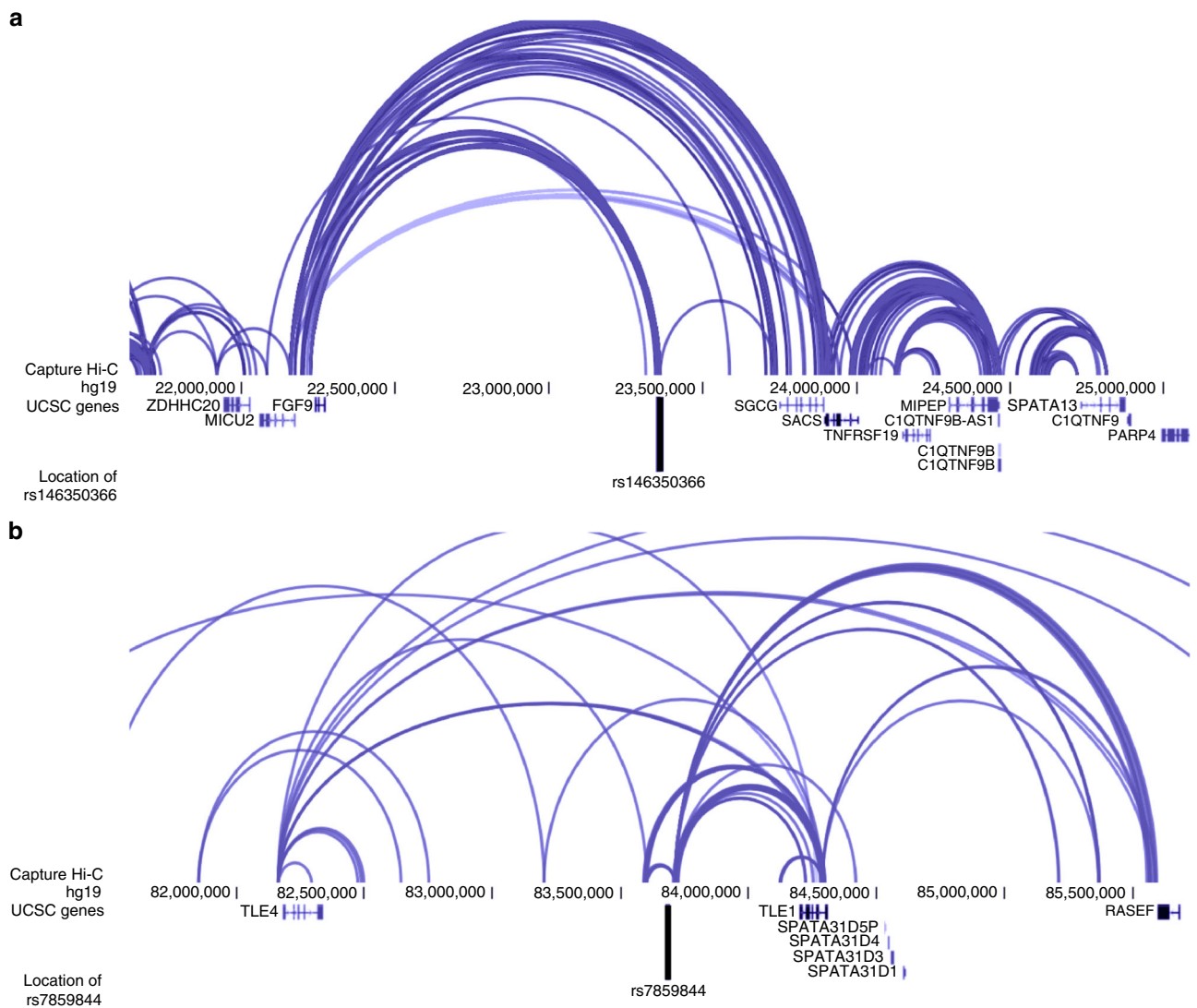

**Fig. 4 Prioritizing genes using chromatin interaction data. a** The GWAS association rs146350366 on chromosome 13 for sporadic miscarriage in the European ancestry meta-analysis shows functional connections to the *FGF9/MICU9* region. The black vertical line represents the location of the signal from GWAS meta-analysis. **b** The GWAS association rs7859844 on chromosome 9 for multiple consecutive miscarriage meta-analysis shows functional connections to the *TLE1* region. The black vertical line represents the location of the signal from GWAS meta-analysis. The 3D Genome Browser[42] was used for data visualization and the endothelial progenitors and fetal thymus data were selected to illustrate the chromatin architecture at loci in panels **a** and **b**, respectively. The visual representations do not infer/confirm possible target tissues for the association.

## Discussion

In this study, we quantify the heritability of miscarriage and identify four distinct susceptibility loci for sporadic and multiple consecutive miscarriage. We acknowledge the fact that the definition we used for multiple consecutive miscarriage differs from that currently used for recurrent miscarriage[1]. Our rationale behind using a definition of at least three consecutive miscarriages (which was also the definition of recurrent miscarriage in much of Europe at the time of the study[2,3]) was to capture the more severe end of the phenotypic spectrum, which would allow to better assess whether recurrent and sporadic events lie on the same phenotypic spectrum. Our findings that the used sporadic miscarriage definition is genetically (and phenotypically) correlated with number of children, whereas the definition used for extracting cases with three or more consecutive miscarriages from self-reported data that resulted in a group that had less children and showed correlations with infertility diagnosis confirms that the used phenotype definitions captured the contrasting ends of the phenotypic spectrum. Hopefully our study paves the way for

future similar studies into the genetics of miscarriage which could provide further evidence for more informed classification of this phenotype.

We found heritability of 29% (95% CI 20–38%) for any miscarriage and a considerably smaller SNP-based heritability for sporadic miscarriage ($h^2 = 1.5\%$ (SE 0.4)). Our study design is limited to interrogate maternal contribution to the genetic architecture of the trait, and it is likely that paternal and fetal contributions are responsible for a proportion of the heritability gap. This also prevents us from investigating the parent–offspring interaction and environmental effects, which have been shown for pre-eclampsia[59]. Overall, it might be expected that genetic factors increasing susceptibility to miscarriage are under negative selection due to their impact on reproductive fitness and hence these will be rare. Up to two-thirds of miscarriages are unrecognized and/or undiagnosed, particularly early miscarriages[60], and thus 'control' women will include some misclassified as not having experienced a miscarriage. This would be expected to attenuate results towards the null and means larger numbers may be

required to accurately quantify SNP-heritability and identify genome-wide significant SNPs; this is likely to affect sporadic miscarriage more so than recurrent miscarriage.

Our results confirm a partly genetic basis to both sporadic and multiple consecutive miscarriage, that does not seem to overlap. Mapping of potential candidate genes at associated loci identified several genes (*FGF9, TLE1, TLE4, E2F8, SIK1*) with a plausible biological role in placental biology and miscarriage etiopathogenesis. However, the involvement of other transcripts at these loci cannot be ruled out and further functional studies are needed. Similarly to other traits, our larger GWAS study fails to replicate findings from previous smaller candidate gene studies[13,15,22], underlining the importance for future larger collaborations to jointly dissect the genetic background of miscarriage.

The MR analyses suggest that smoking may causally increase the risk of sporadic miscarriage, as has also been suggested by epidemiological studies;[26] however, this needs to be validated in independent datasets and the effect of horizontal pleiotropy cannot be ruled out. Finally, our analysis of health outcomes associated with miscarriage confirms previous observations and identifies several novel ones. For some of these diagnoses, including irritable bowel syndrome, asthma, endometrial cancer, self-harm, and ectopic pregnancy, similar epidemiologic associations have been reported previously[61–65], warranting further investigation into underlying mechanisms.

In conclusion, our results show that miscarriage etiopathogenesis is partly driven by genetic variation potentially related to placental biology, and illustrate the utility of large-scale biobank data for understanding this pregnancy complication.

## Methods

**Study cohorts**. Descriptive statistics of the cohorts included in the sporadic and multiple consecutive miscarriage GWAS meta-analyses are presented in Supplementary Data 1 and Supplementary Methods. Our analysis included 69,054 sporadic miscarriage cases, 750 multiple consecutive miscarriage cases and up to 359,469 controls. All individuals gave informed consent at enrollment and recruitment was monitored by relevant institutional ethics review boards.

**Case definitions**. Depending on the type of data available in individual cohorts, miscarriage cases were identified as follows:

*Sporadic miscarriage*: 1 or 2 self-reported miscarriages, or ICD-10 codes O02.1 and O03 on 1 or 2 separate time-points (at least 90 days between episodes). As a result, 26,044 sporadic miscarriage cases were identified from cohorts using only self-report data, 1231 from cohorts using only electronic health record data, and 41,779 from cohorts where both self-reported and EHR data were available.

*Multiple consecutive miscarriage*: (i) five or more self-reported miscarriages, one live birth, no pregnancy terminations, (ii) three or more self-reported miscarriages, no live births, no pregnancy terminations, or (iii) three or more consecutive miscarriages. The first two criteria were used to ensure the consecutive nature of the miscarriages; and (iv) ICD-10 diagnosis code N96.

For a more detailed explanation why these definitions were chosen, please see Supplementary Note 1, Supplementary Fig. 1 and Supplementary Table 1.

Where data allowed, we applied the following exclusion criteria to all cohorts (the aim of these exclusions was to mainly examine associations with idiopathic miscarriage cases and thereby increase the homogeneity of the analyzed phenotype):

- women with early or late menarche (<9 or >17 years), which could indicate underlying hormonal abnormalities
- women with any of diagnoses for conditions associated with increased susceptibility to miscarriage (maternal chromosomal abnormalities, thyroid conditions, neoplasms affecting endocrine glands, thrombophilias, disorders affecting the endocrine system, congenital malformations of genital organs)

**GWAS genotyping and imputation**. Details on cohort-level genotyping, quality control (QC), and imputation can be found in Supplementary Data 2.

**Association analyses**. Details on how association analyses were carried out on the cohort level can be found in Supplementary Data 2. Cohort-level association analyses had been performed using genotype data imputed to suitable reference panels and adjusted for year of birth. Where available and appropriate, additional cohort-specific covariates, such as principal components or genotyping array, were used to correct for potential within-cohort stratification.

**Meta-analysis**. Central QC was conducted using EasyQC[66]. During central QC, allele frequencies and alignment were compared against suitable reference datasets (Haplotype Reference Consortium[67], 1000 Genomes[68]) to detect potential strand issues or large allele frequency deviations from the reference population. Monomorphic markers, and also markers with strand mismatch, poor imputation quality (INFO score < 0.4) or an arbitrary minor allele count cut-off ≤ 6 were excluded from each study prior to the meta-analysis. The results from individual cohorts were meta-analyzed in parallel by two different analysts. All genome-wide significant variants that passed the applied filters (see below) are listed in Supplementary Data 3.

For the trans-ethnic meta-analysis, we used the MR-MEGA software[19], adjusting for the first two principal components. After the meta-analysis, we applied an additional filter for variants present in at least half ($n = 11$) of the cohorts, to rule out spurious associations. This resulted in 8,664,066 variants, and a genome-wide significant association on chromosome 7 (rs10270417). Indels were not considered due to their lower quality. A closer inspection of the effect sizes for the observed association in individual cohorts revealed the association was mainly driven by one of the Chinese-ancestry cohorts (Supplementary Fig. 2c) where the MAF was 0.04%. It is known that BOLT-LMM, used for analysis in the Kadoorie cohort, can overestimate significance for rare SNPs (MAF < 1%) if the case fraction is <10%[20]; therefore we performed additional analyses. The Kadoorie samples have been collected from 10 region centers, therefore we checked for batch effects. Adding batch ID as a covariate did not have a significant impact on the association statistics (original *P*-value $5.0 \times 10^{-10}$, after adding batch ID as covariate $P = 2.2 \times 10^{-15}$). To check possible confounding effect from samples being collected from 10 different region centers, we performed separate analyses for each region center, followed by meta-analysis. As the SNP is very rare, SNPTEST failed to converge in five research center datasets; however, fixed effect meta-analysis detected a similar effect direction in the remaining datasets, although with significant differences in effect magnitude across different region centers ($P_{meta} = 4.8 \times 10^{-4}$; $P_{het} = 5.2 \times 10^{-5}$) and a considerably larger *P*-value compared to the BOLT-LMM results. Although the two methods (SNPTEST and BOLT-LMM) are not directly comparable, given the absence of this variant in other Chinese ancestry cohorts (BGI and UKBB$_{CHI}$), the rs10270417 signal was not taken further for functional annotation.

European-ancestry only sporadic miscarriage meta-analysis was carried out with METAL[21] using inverse variance fixed effects meta-analysis and single genomic correction. Multiple consecutive miscarriage meta-analysis was conducted using METAL[21] and Stouffer's (*P*-value-based effective sample size weighted) method and single genomic correction. After the analysis, sporadic miscarriage European ancestry meta-analysis results were additionally filtered to exclude markers not present in at least half of the cohorts ($n = 7$). From the multiple consecutive miscarriage meta-analysis results we excluded variants that were not present in at least two cohorts, had an average MAF of <0.5%, and a MAF of <0.1% in any of the three cohorts. From the genome-wide significant signals we selected those with the same direction of effect in all three cohorts for further consideration. Indels were not considered due to their lower quality. The quantile–quantile plots, Manhattan plots, and locus zoom plots of the meta-analyses are shown in Figs. 2 and 3 and in Supplementary Figs. 3 and 4.

**Look-up of variants previously associated with recurrent miscarriage**. We conducted a lookup in our summary statistics of all variants included in a recent, extensive and systematic review/meta-analysis of published genetic association studies in idiopathic recurrent spontaneous abortion[15]. The results of this lookup are given in Supplementary Data 4.

**Heritability analysis**. The sporadic miscarriage GWAS European-ancestry meta-analysis summary statistics and LDSC method[23] were used for heritability estimation. The linkage disequilibrium (LD) estimates from European ancestry samples in the 1000 Genomes projects were used as a reference. Heritability estimates were converted to the liability scale using a population prevalence of 0.2 for sporadic miscarriage. Using the UKBB SNP-Heritability Browser (https://nealelab.github.io/UKBB_ldsc/h2_browser.html), we also did a look-up for different versions of the miscarriage phenotype or related phenotypes in the UKBB dataset and observed similar heritability estimates for 'Ever had stillbirth, spontaneous miscarriage or termination' ($h^2 = 0.04$; s.e. = 0.007; population prevalence 31.7%) and 'Number of spontaneous miscarriages' ($h^2 = 0.03$; s.e. = 0.01).

Data from 1853 complete female MZ and 1177 DZ twin pairs and 2268 women from incomplete or opposite sex twin pairs (mean year of birth 1954, range 1893–1989) from the QIMR dataset were used to estimate heritability under a classical twin model, using a multifactorial threshold model in which discrete traits are assumed to reflect an underlying normal distribution of liability (or predisposition). Liability, which represents the sum of all the multifactorial effects, is assumed to reflect the combined effects of a large number of genes and environmental factors each of small effect[69]. All data analyses were conducted using maximum-likelihood analyses of raw data within Mx[70]. Corrections for year

of birth were included with the model, such that the trait value for individual $j$ from family $i$ was parameterized as $x_{ij} = \beta_{age} + \mu$. The phenotypic data, which were constrained to unity, were parameterized as $\sigma^2 = \sigma_A^2 + \sigma_D^2 + \sigma_E^2$ or, $\sigma^2 = \sigma_A^2 + \sigma_C^2 + \sigma_E^2$, where, $\sigma_A^2$ represents additive genetic effect (A); $\sigma_D^2$ represents non-additive genetic effects (D); $\sigma_C^2$ represents shared environmental effects (C) and $\sigma_E^2$ represents non-shared or unique environmental effects (E). The covariance terms were parameterized as $Cov_{MZs} = \sigma_A^2 + \sigma_D^2$ or $\sigma_A^2 + \sigma_C^2$ and $Cov_{DZs} = 0.5\sigma_A^2 + 0.25\sigma_D^2$ or $0.5\sigma_A^2 + \sigma_C^2$.

The significance of variance components was tested by comparing the fit (minus twice the log-likelihood) of the full model which included the effect to that of a nested model in which the effect had been dropped from the model. The difference in log-likelihoods follows an asymptotic chi-square distribution with the degrees of freedom equal to the difference in estimated parameters between the two models. The results of these analyses are summarized in Supplementary Data 5.

**Genetic correlation analyses.** The LDSC method[23] implemented in LD-Hub (http://ldsc.broadinstitute.org)[24] was used for testing genetic correlations between sporadic miscarriage and 760 traits (spanning reproductive, anthropometric, psychiatric, aging, hematological, cardiometabolic, autoimmune, hormone, cancer, smoking behavior, and traits from the UKBB), using the sporadic miscarriage European-ancestry only GWAS meta-analysis summary statistics and data available within the LD Hub resource. False discovery rate (FDR) correction (calculated using the p.adjust function in R) was used to account for multiple testing. Results of the analysis are presented in Supplementary Data 6.

**Associated phenotypes analysis.** The UKBB has extensive phenotype data for 500K individuals, of which 273,465 are women. After applying the same QC criteria that were applied to the subset of individuals used in the genetic association analysis, extensive phenotype data was available for 220,804 women. Among these there were 39,411 sporadic miscarriage cases and 133,545 controls, and 458 multiple consecutive miscarriage cases and 133,675 controls (defined using the criteria applied to define cases and controls for the association analysis). We explored differences in the prevalence of diseases between sporadic miscarriage cases and controls and between multiple consecutive miscarriage cases and controls. Diseases were identified from the UKBB linked Hospital Episode Statistics which provide ICD10 diagnosis codes (UKBB data fields 41202 and 41204). First, for each hospital diagnosis observed among the cases (defined by an ICD10 code; $n = 6840$ for sporadic and $n = 1323$ for multiple consecutive miscarriage, respectively; excluding those used to define the cases), we tested the difference in the proportion of cases with the diagnosis with that among the controls (using a two-sample test for difference in proportions when the number of "successes" and "failures" are greater or equal to five for both populations; otherwise, Fisher's exact test was applied instead). A false discovery rate (FDR) multiple testing correction at level 10% was applied to the P-values. For graphical representation (Supplementary Fig. 5), diagnosis codes were grouped and colored by ICD10 chapters as follows: blood ("Diseases of the blood and blood-forming organs and certain disorders involving the immune mechanism", D50-D89), circulatory ("Diseases of the circulatory system", I00-I99), congenital ("Congenital malformations, deformations and chromosomal abnormalities", Q00-Q99), digestive ("Diseases of the digestive system", K00-K93), ears ("Diseases of the ear and mastoid process", H60-H95), endocrine ("Endocrine, nutritional and metabolic diseases", E00-E90), external ("External causes of morbidity and mortality", V01-Y98), eyes ("Diseases of the eye and adnexa", H00-H59), genitourinary ("Diseases of the genitourinary system", N00-N99), infection ("Certain infectious and parasitic diseases", A00-B99), injury ("Injury, poisoning and certain other consequences of external causes", S00-T98), musculoskeletal ("Diseases of the musculoskeletal system and connective tissue", M00-M99), neoplasms ("Neoplasms", C00-D48), nervous ("Diseases of the nervous system", G00-G99), other ("Factors influencing health status and contact with health services", U04-Z99), perinatal ("Certain conditions originating in the perinatal period", P00-P96), pregnancy ("Pregnancy, childbirth and the puerperium", O00-O99), psychiatric ("Mental and behavioral disorders", F00-F99), respiratory ("Diseases of the respiratory system", J00-J99), skin ("Diseases of the skin and subcutaneous tissue", L00-L99), symptoms ("Symptoms, signs and abnormal clinical and laboratory findings, not elsewhere classified", R00-R99). To rule out the confounding effect of woman's age and parity, we then conducted multivariate logistic regression adjusted for age and number of children for any diseases for which there was statistical evidence of a difference between cases and controls, and applied FDR 5% correction to the P-values (Supplementary Data 7 and 8).

**MR analyses.** We used a two-sample MR approach[29] to explore the possible causal effect of smoking, alcohol, and BMI on sporadic miscarriage. Summary statistics for alcohol use (drinks per week) and smoking initiation (ever smoked regularly) were obtained from the most recent GWAS for these traits[30] and for BMI were obtained from Locke et al. (2015)[31]. Summary associations between each variant and sporadic miscarriage were obtained from the European-ancestry meta-analysis filtered for variants present in at least half of the cohorts. We harmonized the

summary datasets and used IVW[32], WM[33], and MR-Egger[34] methods to obtain a pooled estimate of the associations of genetic variants for smoking, alcohol, and BMI with sporadic miscarriage. Including only variants that reached genome-wide significance in the exposure GWAS in our MR analyses, 369/378 variants associated with smoking, 96/99 variants associated with alcohol use, and 97/97 variants associated with BMI were present in our European-ancestry sporadic miscarriage GWAS. The results of these analyses are presented in Supplementary Data 9.

We conducted a (hypothesis generating) MR phenome-wide (PheWAS) analysis of multiple consecutive miscarriage (using a per allele genetic risk score from the four GWAS significant SNPs; rs7859844, rs143445068, rs138993181, rs183453668) in relation to 17,037 outcomes using the PHESANT[35] package in UKBB ($n = 168,763$) (Supplementary Fig. 7). The analysis was adjusted for year of birth and the top 10 PCs. Overall, the MR-PheWAS did not show any evidence of causal effects (Supplementary Fig. 8). Only three outcomes reached Bonferroni-corrected levels of statistical significance ($P < 2.93 \times 10^{-6}$), including one outcome related to alcoholism and one related to post-traumatic stress disorder. However, both of these were single items from instruments that included 11 items (alcohol use questionnaire) and 21 items (post-traumatic/traumatic event questionnaire), respectively, with none of the other items reaching suggestive thresholds of statistical significance. The third outcome to show association below this P-value threshold was a job coding (scenery designer or costume designer) that is one of which lies in 42-item employment history category (MR analyses did not suggest effects on any other jobs in this list).

**Functional annotation.** The FUMA platform designed for prioritization, annotation, and interpretation of GWAS results[41] was used for functional annotation of association signals from the GWAS meta-analyses (Supplementary Methods).

To narrow down potential candidate genes, we first extracted SNPs in high LD ($r^2 > / = 0.8$) with the lead signals and overlapped those with chromatin data (ChIPseq for histone modifications and DHS for chromatin accessibility, both indicative of regulatory elements) using HaploReg. We then used Hi-C chromatin interaction datasets to visualize topologically associated domains (TADs) in the region and Capture Hi-C data for various tissues to further explore interactions within the TAD domain. Data was visualized using the 3D Genome Browser[42] (http://3dgenome.org) using the datasets available via the browser[71,72]. TADs are relatively conserved across different tissue types and define the boundaries for potential genomic interactions[73]. In the main text (Fig. 4), visual representations are shown for single tissues to show representative signals at a given locus and to illustrate chromatin architecture (promoter–enhancer interactions) at the locus, to infer possible target genes(s). The visual representations do not infer/confirm possible target tissues for the association.

For sporadic miscarriage, we used the summary statistics of our EUR-ancestry meta-analysis. A total of five candidate SNPs were identified ($r^2 \geq 0.6$ with rs146350366) in the associated locus on chr13, all of them intergenic (Supplementary Data 11). Of these, rs188519103, located 6.9 kb 5′ of SNORD36 had the lowest RegulomeDB score (4—evidence of transcription factor binding and DNase peak). Potential candidate genes were mapped using eQTL and chromatin interaction data (Supplementary Data 12).

In the multiple consecutive miscarriage analysis, we had three associated loci with consistent effect direction in all three cohorts. For the signal on chromosome 9, 53 candidate SNPs were identified by FUMA (Supplementary Data 13). Of these, rs12004880 had a RegulomeDB score of 3a ("TF binding + any motif + DNase peak"), while four SNPs had a CADD score of >12.37, indicating potential pathogenicity[74]. A total of 50 candidate genes were proposed (Supplementary Data 14), among them protein-coding TLE1, TLE4, PSAT1, IDNK, GNAQ, RASEF, SPATA31D1, and FRMD3. On chromosome 11, rs143445068 (RegulomeDB score 3a) and rs140847838 were highlighted as potential candidate SNPs in the associated region located in the intron of NAV2. Chromatin interaction mapping proposed another 17 candidate genes, including DBX1, HTATIP2, E2F8, ZDHHC13, MRGPRX2. Hi-C map in ovaries from the 3D Genome Browser[42] is shown in Supplementary Fig. 10. Finally, for association signal on chromosome 21, no other candidate SNPs in addition to the lead signal rs183453668 were identified, and a total of 10 candidate genes were suggested by chromatin interaction data. Hi-C map in ovaries and Capture Hi-C data visualization in endothelial progenitors are shown on Supplementary Fig. 11.

To further narrow down the list of potential candidate genes, we used measures of constraint (pLI score[36] and oe ratio) from gnomAD v2.1.1[37] and data on mouse phenotypes from the International Mouse Phenotype Consortium[39] (https://www.mousephenotype.org) and the Mouse Genome Informatics database[38] (http://www.informatics.jax.org/phenotypes.shtml). In brief, the pLI score reflects the probability that a given gene is LoF intolerant and pLI scores ≥0.9 reflect extreme intolerance to protein-truncating variation, which could point to a reproductive disadvantage in heterozygous LoF cases[36]. The oe ratio is the ratio of the oe number of LoF variants in that gene. It is a continuous measure, where lower oe values indicate the gene is under stronger selection for that kind of variation compared to a gene with higher values. The upper bound of the oe 90% CI < 0.35 is recommended for thresholding. In mouse phenotype data, we first and foremost looked at potentially early (embryonic/prenatal/preweaning) lethal phenotypes. We

also compared our candidate genes against a recently published carefully curated list[40] of genes potentially relevant to infertility or recurrent intra-uterine death, which were selected based on scores of genetic constraint, murine phenotypic information, and the cell 'essentialome', i.e. genes, that are essential for human cell lines.

Using this approach, in each locus the most likely candidate genes were considered to be those that either show evidence of evolutionary constraint and are therefore relevant for reproductive fitness or are associated with an early lethal phenotype in mice (Supplementary Data 15). Previous publications showing evidence the gene is linked with either placental function, embryo viability, pregnancy maintenance, or miscarriage/pregnancy loss were also considered when prioritizing potential candidate genes.

**Reporting summary**. Further information on research design is available in the Nature Research Reporting Summary linked to this article.

## Data availability

The GWAS meta-analysis summary statistics that support the findings of this study are available for download from http://www.geenivaramu.ee/tools/misc_sumstats.zip . The analyses in this manuscript included data from UK Biobank, http://www.ukbiobank.ac.uk/, under applications 17805, 11867, and 16729; Estonian Biobank, https://www.geenivaramu.ee/en; ALSPAC (http://www.bristol.ac.uk/alspac/); China Kadoorie Biobank (http://www.ckbiobank.org/). All QC and GWAS meta-analyses were carried out with standard tools and pipelines. The analyses in this paper also use data from the 3D Genome Browser, http://promoter.bx.psu.edu/hi-c/; GTEx, https://gtexportal.org/home/; International Mouse Phenotype Consortium, https://www.mousephenotype.org; Mouse Genome Informatics database; http://www.informatics.jax.org/phenotypes.shtml; GWAS atlas, https://atlas.ctglab.nl.

## Code availability

Cohort-level analyses were carried out with SNPTEST v2.5.0, BOLT-LMM v2.3.2, EPACTS 3,3, plink 1.9, RAREMETALWORKER, Mach2dat. Before central meta-analysis, data quality control was conducted using Easy QC software (v17.6). Central meta-analysis was conducted using the MR-MEGA (0.1.5; https://genomics.ut.ee/en/tools/mr-mega) and METAL (version released on 2011-03-25; https://genome.sph.umich.edu/wiki/METAL_Documentation) software. Follow-up analyses were in part carried out using FUMA (1.3.1; http://fuma.ctglab.nl/), using data from ANNOVAR (17-07-2017) and GWAS catalog (e91_r2018-02-06). Gene-based testing was carried out with MAGMA 1.06 implemented in FUMA. Functional annotations of variants were obtained from HaploReg 4.1. SNP heritability and genetic correlations were calculated using LDSC software (1.0.0) and LDSC software implemented in LD-Hub (http://ldsc.broadinstitute.org/), respectively. Colocalization analyses were performed using R 3.4.3 and 3.5.1 (https://www.r-project.org/), the coloc R package and the coloc.abf() function and LocusCompareR (http://www.locuscompare.com/) was used for visualization. MR-PheWAS was conducted using the PHESANT R package. Fig. 1 was created using the maps package in R. All other analyses were carried out in the R environment.

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

## Acknowledgements

T.L. is supported by European Commission Horizon 2020 research and innovation program (project WIDENLIFE, grant number 692065); Estonian Ministry of Education and Research (grants IUT34-16, PUT IUT20-60, PUTJD726, and MOBTP155); Enterprise Estonia (grant EU49695). J.C.C. is funded by the Oxford Medical Research Council Doctoral Training Partnership (Oxford MRC DTP) and the Nuffield Department of Clinical Medicine, University of Oxford. S.L. is supported by a Novo Nordisk Postdoctoral Fellowship run in partnership with the University of Oxford. C.M.L. is supported by the Li Ka Shing Foundation, WT-SSI/John Fell funds, Oxford, NIHR Oxford Biomedical Research Centre, Oxford, Widenlife and NIH (5P50HD028138-27). D.F.C. is supported by grants from the National Institutes of Health (R01HD078641, R01MH101810, and P51OD011092). R.M. is funded by Estonian Ministry of Education and Research (grant PRG687). T.F. is supported by the NIHR Biomedical Research Centre, Oxford. S.E.M. and J.N.P. were supported by NHMRC applications APP1103623 and APP1084325. Partners HealthCare Biobank is supported by NHGRI grant U01HG008685. This study was funded by EU H2020 grant 692145, Estonian Research Council Grant IUT20-60, IUT24-6, and European Union through the European Regional Development Fund Project No. 2014-2020.4.01.15-0012 GENTRANSMED and 2014-2020.4.01.16-0125. Data analyses were carried out in part in the High-Performance Computing Center of University of Tartu. This study was funded by The Lundbeck Foundation, Denmark. This research has been conducted using the Danish National Biobank resource, supported by the Novo Nordisk Foundation. Computation used the Oxford Biomedical Research Computing (BMRC) facility, a joint development between the Wellcome Centre for Human Genetics and the Big Data Institute supported by Health Data Research UK and the NIHR Oxford Biomedical Research Centre. Financial support was provided by the Wellcome Trust Core Award Grant Number 203141/Z/16/Z. The views expressed are those of the author(s) and not necessarily those of the NHS, the NIHR or the Department of Health. We thank the study subjects for their valuable participation. We thank the twins and their families for their participation in the QIMR study. We acknowledge with appreciation all women who participated in the QIMR endometriosis study. We are extremely grateful to all the families who took part in the ALSPAC study, the midwives for their help in recruiting them, and the whole ALSPAC team, which includes interviewers, computer and laboratory technicians, clerical workers, research scientists, volunteers, managers, receptionists and nurses. The authors wish to acknowledge the services of the Lifelines Cohort Study, the contributing research centers delivering data to Lifelines, and all the study participants. A full list of acknowledgements is provided in the Supplementary Note 2.

## Author contributions

T.L., A.L.G.S, T.F., J.N.P., J.C.C, S.La., J.B., C-Y.C., K.L., S.Liu, and A.R. carried out data analysis; H.S., Z.C., D.F.C., B.J., L.L., N.G.M., B.N., R.N., R.G.W., S.E.M., R.M., D.A.L., and C.M.L. coordinated cohort-level analyses; M.L., I.Y.M., J.S., M.S.A., L.Y., C.B., S.D.G., J.B-G., Ø.H., D.M.H., X.J., S.J., J.J., C.K., V.K., P.A.L., A.D.B., O.M., M.N., T.W., A.M., G. W.M., A.P.M., P.B.M., P.R.N., D.R.N., M.L., S.S., A.S., K.Z., I.G.; and D.A.L. provided and/or processed (phenotype) data; T.L., A.L.G.S., R.M., D.A.L., and C.M.L. drafted the manuscript; All authors contributed to the final version of the manuscript.;

## Competing interests

D.A.L. has received support from Roche Diagnostics and Medtronic Ltd for biomarker research unrelated to that presented here. The other authors have no competing interests.

## Additional information

Triin Laisk [1,2,3,48 ✉], Ana Luiza G. Soares [4,5,48], Teresa Ferreira[6,48], Jodie N. Painter[7,48], Jenny C. Censin[6,8], Samantha Laber[6,8], Jonas Bacelis [9], Chia-Yen Chen[10,11,12], Maarja Lepamets[2,13], Kuang Lin[14], Siyang Liu[15,16], Iona Y. Millwood[14,17], Avinash Ramu[18], Jennifer Southcombe[19], Marianne S. Andersen[20], Ling Yang[14,17], Christian M. Becker [19], Anders D. Børglum [21,22,23], Scott D. Gordon [7], Jonas Bybjerg-Grauholm [21,24], Øyvind Helgeland[25,26], David M. Hougaard [21,24], Xin Jin[15,27], Stefan Johansson [26,28], Julius Juodakis [29], Christiana Kartsonaki[14,17], Viktorija Kukushkina[2,13], Penelope A. Lind [7], Andres Metspalu [2], Grant W. Montgomery [30], Andrew P. Morris [2,8,31], Ole Mors[21,32], Preben B. Mortensen[21,33], Pål R. Njølstad [26,34], Merete Nordentoft[21,35], Dale R. Nyholt [36], Margaret Lippincott[37], Stephanie Seminara[37], Andres Salumets[1,3,38,39], Harold Snieder [40], Krina Zondervan [8,19], Thomas Werge [21,41,42], Zhengming Chen [14], Donald F. Conrad[18], Bo Jacobsson [9,25], Liming Li [43], Nicholas G. Martin [7], Benjamin M. Neale [10,11,12], Rasmus Nielsen [44,45], Robin G. Walters [14,17], Ingrid Granne[19,49], Sarah E. Medland [7,49], Reedik Mägi[2,49], Deborah A. Lawlor [4,5,46,49] & Cecilia M. Lindgren [6,8,47,49 ✉]

[1]Department of Obstetrics and Gynecology, Institute of Clinical Medicine, University of Tartu, Tartu, Estonia. [2]Estonian Genome Center, Institute of Genomics, University of Tartu, Tartu, Estonia. [3]Competence Centre on Health Technologies, Tartu, Estonia. [4]MRC Integrated Epidemiology Unit at the University of Bristol, Bristol, UK. [5]Population Health Sciences, Bristol Medical School, University of Bristol, Bristol, UK. [6]Big Data Institute, Li Ka Shing Center for Health for Health Information and Discovery, Oxford University, Oxford, UK. [7]QIMR Berghofer Medical Research Institute, Herston, QLD, Australia. [8]Wellcome Centre for Human Genetics, University of Oxford, Oxford, UK. [9]Department of Obstetrics and Gynecology, Sahlgrenska University Hospital Östra, Gothenburg, Sweden. [10]Analytic and Translational Genetics Unit, Massachusetts General Hospital, Boston, MA, USA. [11]Psychiatric and Neurodevelopmental Genetics Unit, Massachusetts General Hospital, Boston, MA, USA. [12]Broad Institute of MIT and Harvard, Cambridge, MA, USA. [13]Institute of Molecular and Cell Biology, University of Tartu, Tartu, Estonia. [14]Clinical Trial Service Unit & Epidemiological Studies Unit (CTSU), Nuffield Department of Population Health, University of Oxford, Oxford, UK. [15]BGI-Shenzhen, Shenzhen, 518083 Guangdong, China. [16]Bioinformatics Centre, Department of Biology, University of Copenhagen, 2200 Copenhagen, Denmark. [17]Medical Research Council Population Health Research Unit (PHRU), University of Oxford, Oxford, UK. [18]Department of Genetics, Washington University in St. Louis, Saint Louis, MO, USA. [19]Nuffield Department of Women's and Reproductive Health, University of Oxford, Oxford, UK. [20]Department of Endocrinology, Odense University Hospital, Odense, Denmark. [21]iPSYCH, The Lundbeck Foundation Initiative for Integrative Psychiatric Research, Aarhus, Denmark. [22]Department of Biomedicine and Center for Integrative Sequencing, iSEQ, Aarhus University, Aarhus, Denmark. [23]Center for Genomics and Personalized Medicine, Aarhus University and University Hospital, Aarhus, Denmark. [24]Department for Congenital Disorders, Statens Serum Institut, Copenhagen, Denmark. [25]Department of Genetics and Bioinformatics, Health Data and Digitalisation, Norwegian Institute of Public Health, Oslo, Norway. [26]KG Jebsen Center for Diabetes Research, Department of Clinical Science, University of Bergen, N-5020 Bergen, Norway. [27]School of Medicine, South China University of Technology, Guangzhou, 510006 Guangdong, China. [28]Department of Medical Genetics, Haukeland University Hospital, N-5021 Bergen, Norway. [29]Department of Obstetrics and Gynecology, Institute of Clinical Sciences, Sahlgrenska Academy, University of Gothenburg, Gothenburg, Sweden. [30]University of Queensland, St Lucia, QLD, Australia. [31]Department of Biostatistics, University of Liverpool, Liverpool, UK. [32]Psychosis Research Unit, Aarhus University Hospital – Psychiatry, Aarhus, Denmark. [33]National Centre for Register-Based Research, Aarhus University, Aarhus, Denmark. [34]Department of Pediatrics, Haukeland University Hospital, Bergen, Norway. [35]Copenhagen University Hospital, Mental Health Center Copenhagen, Mental Health Services in the Capital Region of Denmark, Copenhagen, Denmark. [36]School of Biomedical Sciences, Faculty of Health, Queensland University of Technology, Brisbane, QLD, Australia. [37]Reproductive Endocrine Unit, Massachusetts General Hospital, Boston, MA, USA. [38]Institute of Bio- and Translational Medicine, University of Tartu, Tartu, Estonia. [39]Department of Obstetrics and Gynecology, University of Helsinki and Helsinki University Hospital, Helsinki, Finland. [40]Department of Epidemiology, University of Groningen, University Medical Center Groningen, Groningen, the Netherlands. [41]Institute of Biological Psychiatry, MHC

Sct. Hans, Mental Health Services Copenhagen, Roskilde, Denmark. [42]Department of Clinical Medicine, University of Copenhagen, Copenhagen, Denmark. [43]Department of Epidemiology & Biostatistics, Peking University Health Science Centre, Peking University, Beijing, China. [44]Department of Integrative Biology, University of California Berkeley, Berkeley, CA, USA. [45]Centre for GeoGenetics, Natural History Museum of Denmark, University of Copenhagen, Copenhagen, Denmark. [46]Bristol National Institute of Health Research Biomedical Research Centre, Bristol, UK. [47]Program in Medical and Population Genetics, Broad Institute, Boston, MA, USA. [48]These authors contributed equally: Triin Laisk, Ana Luiza G. Soares, Teresa Ferreira, Jodie N. Painter. [49]These authors jointly supervised this work: Ingrid Granne, Sarah E. Medland, Reedik Mägi, Deborah A. Lawlor, Cecilia M. Lindgren. ✉email: triin.laisk@ut.ee; cecilia.lindgren@bdi.ox.ac.uk

