## [Peer Review File · Nature Communications]

Reviewers' comments:

Reviewer #1 (Remarks to the Author):

1. The authors perform the 1st GWAS for miscarriage. Essentially, they find only 1 significant signal for sporadic miscarriage and 3 signals for habitual miscarriage in a much smaller analysis of 750 cases, both only among Europeans. Unfortunately linkage to plausible underlying genes is negative using tissue expression data and so is informed only by Hi-C data which is poorly specific, i.e. the one signal for sporadic events is linked to 47 genes (why do they choose the data specifically for endothelial progenitors?) and the 3 signals for habitual events are linked to 50 genes. Hence linkage to possible biological mechanisms is hugely speculative and should be omitted from the Abstract. The conclusion that the findings 'implicate novel biology' is incorrect.
2. Abstract "biobank-scale....analyses". 'Biobank' implies a large study sample size. Instead, here they mean a large number of variables, often termed 'phenome-wide'.
3. Page 6, Methods. It is odd that they use ICD-10 diagnosis code N96 for habitual abortion, which is defined as having 2 or more abortions, but choose to combine this with 3 or more self-reported miscarriages. It is estimated that N96 affects ~1% of women. How frequent were self-reported 2+ and 3+ events and in which sample?
4. Page 13, of the previous 333 reported variants from candidate gene studies, did any pass a threshold of $p < 0.05/333$? Comment that fewer than expected by chance (5%) were nominally significant.
5. Page 13, unfortunately it appears that the QIMR twin study does not have cases of habitual miscarriage to assess whether this condition is more heritable than the sporadic event, as might be suspected by the differing numbers of GWAS signals.
6. What is the genetic correlation between sporadic and habitual miscarriage? What is the association of the sporadic event signal on habitual events, and vice versa?
7. Page 15, describes a long section on phenotypic associations. Clarify which of these were or were not confirmed by genetic correlations. For a genetics paper, it seems sensible to focus only on those that also show genetics correlations.
8. Page 16, their phenome-wide MR analysis found no causal associations which is not surprising given the few signals in their instrumental variable (except Fig S6 seems to show 2 significant points?). It would be more appropriate instead to examine MR for miscarriage as the outcome for candidate exposure traits with more powered instruments, such as BMI, diabetes, asthma, mental health etc.
9. Page 16, para 3. "(13% of them protein coding)" Clarify that this is a surprisingly small proportion.

Reviewer #2 (Remarks to the Author):

Laisk and co-authors report the first large-scale GWAS for sporadic (69,054 cases from five ancestries) and recurrent (750 cases of European ancestry) miscarriage (359,469 controls). They identify genome-wide significant associations for sporadic miscarriage ($n=1$, $p=3.2 \times 10^{-8}$ with $OR=1.4$) and recurrent miscarriage ($n=3$, $p=1.3 \times 10^{-8}$ with $OR=1.7$, $p=5.2 \times 10^{-9}$ with $OR=3.4$, $p=2.8 \times 10^{-8}$ with $OR=3.8$). The study is a comprehensive analysis of the genetic architecture of miscarriage with Mendelian randomization, heritability and genetic correlation analyses. Despite the importance of miscarriage in women's health, there are concerns about this study. Rigorous group definitions are fundamental for validity of the findings.

Introduction

p. 5: The authors introduce sporadic miscarriage as 1-2 miscarriages and recurrent as >3 consecutive miscarriages; however, it should be noted that this is not the current definition of sporadic and recurrent miscarriage as supported by clinical guidelines (i.e., ASRM, ESHRE).

Further in the Introduction (p. 6), the authors state that current guidelines define recurrent miscarriage as the loss of >2 pregnancies, which is correct. Also, on p. 6 they state that in the present study they define recurrent miscarriage as having had >3 self-reported miscarriages or code N96 (i.e., >2 miscarriages) such that they will capture the more severe end of the phenotypic distribution to differentiate it from sporadic miscarriage. This is confusing for the reader.

p. 6: The authors state that miscarriage increases with maternal age, which is well known to be associated with increased aneuploidy. Is there any consideration of the U-shaped distribution in maternal age for aneuploidy? Is that relevant in the different populations used in the study? Is there any exclusion, correction or stratification for maternal age? This is fundamental for the analysis.

Results and Discussion

p. 7 Please replace "...on chromosome 7p15.2..." with "...on chromosome 7 in band p15.2...". There is no chromosome 7p15.2.

p. 18 Please replace "...ZDHHC13, MRGPRX2..." with "...ZDHHC13 and MRGPRX2...".

p. 18 Please replace "...HSF2BP, RRP1B..." with "...HSF2BP and RRP1B...".

Methods

p. 20 Is the question posed in the UKBB, "Have you ever had any stillbirths, spontaneous miscarriages or terminations?" problematic. The etiologies of stillbirth could be very different from those resulting in miscarriages, and social abortions (i.e., terminations) could be frequent and not have any underlying genetic risk. This grouping could be very heterogenous and affect the quality of the results. Were social abortions excluded?

p. 20 Please replace "...participants, data has..." with "...participants, data have...".

p. 23 and p. 37 There is a reference to the QIMR Endo samples but no previous mention of an analysis of endometriosis. The authors describe MR for endometriosis and recurrent miscarriage (p. 37) but this is given in the Methods and is not present in the Results and Discussion.

p. 26 Please replace "...sequencing data is..." with "...sequencing data are...".

p. 27 The case definitions continue to be confusing from that stated in the Introduction. The authors define sporadic miscarriage here by two standards (e.g., self-report or two other ICD-10 codes), which potentially results in combining patients with different etiologies. The authors should clearly state the number in each category. Recurrent miscarriage is defined by four different categories including code N96, which can be defined by only 2 miscarriages. Also, what is the relevance in the recurrent classification of requiring miscarriages to be consecutive? If there were an underlying genetic etiology that resulted in an increased risk for chromosomal non-disjunction (e.g., spindle fiber disorder) and there were multiple aneuploid miscarriages (e.g., trisomy 21, trisomy 18, and trisomy 13 as has been observed in single patients) separated by euploid pregnancies, would this not qualify as recurrent miscarriage? What would the results of the study be had sporadic miscarriage been defined as a single miscarriage and recurrent miscarriage as the loss of >2 pregnancies? This is critical and the results are impacted by mixing categories of patients.

p. 28 Is there a bullet point missing before the second indented criterion "women with any of diagnoses for..." under Exclusion criteria.

p. 36 Please replace "...one of a which lies in 42-item..." with "...one of which lies in a 42-item...".

p. 36 Please replace, "The nature of PheWAS are that..." with "The nature of PheWAS is that...".

p. 36 Please replace "...This approach is not able to conclusively determine the..." with "...This approach is not able to determine conclusively the...".

p. 39 The authors describe that they use Hi-C in ovaries but no description is provided of the specific cell line nor from where the data have been obtained. They also refer that for Fig. S9 they use endothelial progenitors without describing the cell line or tissue used. Are the TADs different from standard tissues (e.g., LCLs or fibroblasts)?

p. 40 Please italicize the complete gene symbol *TLE1*.

p. 41 Please insert "an" between "with imputation".

p. 41 Please insert "or" prior to "genotype".

Supplementary Text and Figures

p. 9 and 10 What is Symptomes?

Reviewers' comments:

Please note the Supplementary tables in Excel format have been uploaded as a Supplementary Dataset and are not included in the pdf generated by the manuscript tracking system.

Reviewer #1

1. The authors perform the 1st GWAS for miscarriage. Essentially, they find only 1 significant signal for sporadic miscarriage and 3 signals for habitual miscarriage in a much smaller analysis of 750 cases, both only among Europeans. Unfortunately, linkage to plausible underlying genes is negative using tissue expression data and so is informed only by Hi-C data which is poorly specific, i.e. the one signal for sporadic events is linked to 47 genes (why do they choose the data specifically for endothelial progenitors?) and the 3 signals for habitual events are linked to 50 genes. Hence linkage to possible biological mechanisms is hugely speculative and should be omitted from the Abstract. The conclusion that the findings 'implicate novel biology' is incorrect.

We thank the reviewer for raising the issue of why we chose to exhibit the Hi-C data specifically for endothelial progenitors. We first extracted SNPs in high LD ($r^2 \geq 0.8$) with the tag SNP rs146350366 and overlapped those with chromatin data (ChIPseq for histone modifications and DHS for chromatin accessibility, both indicative of regulatory elements) using HaploReg. The SNPs in this locus fall in open chromatin regions (DHS peaks) of a variety of cell types/tissues, including iPSCs, heart, muscle, lung and liver. When we then used the 3D Genome Browser to look at promoter capture Hi-C at this locus, we observed that the associated region is in contact with the FGF9 promoter in many of the selected tissues, suggesting that the 3D chromatin structure is not specific to a single tissue, but that folding of the associated region and promoter regions of FGF9 is shared between a number of tissues. In the manuscript, we show one tissue to show representative signal at a given locus, and to illustrate the chromatin architecture (promoter-enhancer interactions) at the locus, to infer possible target gene(s), but acknowledge that this is to no means confirming a possible target tissue for the association. This was also true for rs7859844, where SNPs in high LD ($r^2 \geq 0.8$) with the tag SNP overlapped DHSseq peaks in multiple tissues, including lung, skin, breast, brain, heart, and muscle. More screenshots of Promoter Capture Hi-C data from the 3D Genome Browser in available tissue types are attached at the end of this letter for reference. We have now also added a brief explanation to the manuscript and edited figure legends accordingly.

Also, driven by this remark, we reperformed candidate gene mapping and selection at associated loci, using a more complex approach incorporating measures of constraint (pLI scores and observed/expected ratio from gnomAD v2.1.1¹) and data from mouse models based on two databases^{2,3}. We also compared our candidate genes against a recently published carefully curated list of genes potentially relevant to infertility or recurrent intra-uterine death⁴, which were selected based on scores of genetic constraint, murine phenotypic information, and the cell 'essentialome', i.e. genes, that are essential for human cell lines. Using this approach, we believe that the most likely candidate genes in each locus are those that either show evidence of evolutionary constraint and are therefore relevant for reproductive fitness or are associated with an early lethal phenotype in mice. Previous publications showing evidence the gene is linked with either placental function, embryo viability, pregnancy maintenance or miscarriage/pregnancy loss were also considered when prioritizing potential candidate genes. Using this approach, we were able to map most likely candidate genes in all four associated loci, and interestingly, all of the

most likely candidate genes can be linked to placental biology. Given this, we believe we have reduced the speculative nature of our findings and would like to keep this information in the abstract.

2. Abstract "biobank-scale....analyses". 'Biobank' implies a large study sample size. Instead, here they mean a large number of variables, often termed 'phenome-wide'.

In the abstract we have a sentence stating "We further investigate the genetic architecture of miscarriage with biobank-scale Mendelian randomization, heritability, and genetic correlation analyses." While it is true that the Mendelian randomization analyses we carried out were also phenome-wide, with this sentence we wanted to highlight the fact that the mentioned analyses included a large sample size (the SNP-heritability and genetic correlation analyses were conducted using the sporadic miscarriage European ancestry GWAS summary statistics which included data for 49,996 sporadic miscarriage cases and 174,109 female controls) and are therefore indeed biobank-scale.

3. Page 6, Methods. It is odd that they use ICD-10 diagnosis code N96 for habitual abortion, which is defined as having 2 or more abortions, but choose to combine this with 3 or more self-reported miscarriages. It is estimated that N96 affects ~1% of women. How frequent were self-reported 2+ and 3+ events and in which sample?

It is true that according to current ESHRE and ASRM guidelines, recurrent miscarriage/N96 is defined as 2 or more spontaneous abortions. However, this has not always been the case, and the most recent set of ESHRE guidelines was published in November 2017. Before that, the recurrent pregnancy loss definition varied from two miscarriages (not necessarily consecutive) (ASRM Practice Committee, 2013⁵; Zegers-Hochschild et al., 2009⁶), to three or more consecutive pregnancy losses (Jauniaux et al., 2006⁷; RCOG Green Top Guideline, 2011⁸). All the recurrent miscarriage cohorts (UKBB, EGCUT, ALPSAC) included in this study are from Europe and have received the N96 diagnosis before 2018, therefore we believe it is acceptable to combine the N96 with self-reported data.

In the UKBB, which is the largest contributing cohort to the recurrent miscarriage analysis, we identified a total of 458 recurrent miscarriage cases (all ethnicities, including 421 White British cases, who were included in the final GWAS). Of these, 21 had the N96 diagnosis code, while 150 had reported at least five miscarriages, one live birth, no elective terminations (to ensure the consecutiveness of miscarriages) and 291 had reported at least 3 miscarriages, no live births or elective terminations. The UKBB includes 273,472 women, resulting in a prevalence of 0.2% for recurrent miscarriage. In the sporadic miscarriage analysis, a total of 38,709 self-reported cases from different ancestries were identified, 7,739 of them with 2 miscarriages. If we combined these with the 3 and more miscarriage cases, it would result in ~8,200 cases and a prevalence of >3% for recurrent miscarriage. It should be noted that the mean age of UKBB participants is over 50 years, resulting in a lower prevalence of reproductive-health related diagnoses than expected, therefore we believe that prevalence of >3% in this particular cohort is overestimated.

Of note, the ESHRE 2018 guidelines acknowledged that using two miscarriage as the definition of recurrent miscarriage was at least in part to facilitate research, improve shared decision making with patients and provide them psychological support, rather than because of more specific evidence that this defined a unique phenotype. In fact, the ESHRE guidelines acknowledge not all guideline group members agreed with this definition demonstrating the continuing uncertainty in the field regarding the definition of RM. Data support increasing numbers of previous miscarriages as being a risk factor for further miscarriage⁹ and increasing likelihood of aneuploid miscarriage¹⁰. Thus our rationale behind

using a definition of at least 3 miscarriages (which was also the definition of recurrent miscarriage in much of Europe at the time the study was started) was to capture the more severe end of the phenotypic spectrum, which would allow to better assess whether recurrent and sporadic events lie on the same phenotypic spectrum.

In the Estonian Biobank (EGCUT) cohort, we had 113 recurrent miscarriage cases. Due to the high prevalence of elective abortions in this cohort, nearly all cases were identified using the N96 diagnosis code. The EGCUT data freeze used for this analysis included 34,320 women¹¹, resulting in a prevalence of 0.3%. Among the EGCUT sporadic miscarriage cases, 18% (605) had 2 miscarriages. In ALSPAC, recurrent miscarriage cases were identified using only self-reported data, therefore all cases have at least 3 miscarriages. Among the ALSPAC sporadic miscarriage cases, 729 reported having experienced 2 miscarriages. It is worth mentioning that the EGCUT is a population-based biobank similar to UKBB, although the age distribution of the participants is different and many female participants have a much wider age distribution (18-85), whereas the UKBB female participants are older (37-73). The ALSPAC cohort on the other hand represents a prospective pregnancy/birth cohort recruiting women with successful live births, and therefore it is expected that this cohort is more fertile than the population-based cohorts.

4. Page 13, of the previous 333 reported variants from candidate gene studies, did any pass a threshold of $p < 0.05/333$? Comment that fewer than expected by chance (5%) were nominally significant.

For the previously reported variants, the p -values from the current sporadic and recurrent miscarriage GWAS meta-analyses are provided in Supplementary Table 4. None of these variants passed a p -value threshold of 1.5×10^{-4} . We have now added a comment that the number of variants passing the nominal significance threshold ($p < 0.05$) is less than expected by chance (page 13).

5. Page 13, unfortunately it appears that the QIMR twin study does not have cases of habitual miscarriage to assess whether this condition is more heritable than the sporadic event, as might be suspected by the differing numbers of GWAS signals. What is the genetic correlation between sporadic and habitual miscarriage? What is the association of the sporadic event signal on habitual events, and vice versa?

The developers of the LDSC software that was used for heritability estimates in this study recommend using at least 5000 samples for reliable heritability estimates. Unfortunately, the current effective sample size for recurrent miscarriage (3,075) is insufficient for analyzing the genetic correlation between sporadic and recurrent miscarriage robustly.

All the recurrent miscarriage signals were insignificant in the sporadic miscarriage analysis, and the sporadic miscarriage signal had a p -value of 0.04981 (so barely nominally significant) in the recurrent miscarriage analysis.

7. Page 15, describes a long section on phenotypic associations. Clarify which of these were or were not confirmed by genetic correlations. For a genetics paper, it seems sensible to focus only on those that also show genetics correlations.

We agree with the reviewer that in addition to simply testing phenotypic correlations it would be beneficial to test for genetic correlations in parallel. Unfortunately, for most of these diagnoses corresponding sufficiently sized GWAS data are not available (as mentioned above, a meaningful LDSC analysis an effective sample size of >5000). We have now addressed this issue in the manuscript text as well and pointed out those phenotypes for which there was also support from genetic correlation analyses (pages 15-16).

Although the main focus of our paper is indeed the genetics of miscarriage, we have undertaken additional more broader analyses to increase our understanding of this phenotype, which could help make informed decision and choices when planning future (genetic) studies. Therefore, we would prefer to keep the associated phenotypes section in the manuscript.

8. Page 16, their phenome-wide MR analysis found no causal associations which is not surprising given the few signals in their instrumental variable (except Fig S6 seems to show 2 significant points?). It would be more appropriate instead to examine MR for miscarriage as the outcome for candidate exposure traits with more powered instruments, such as BMI, diabetes, asthma, mental health etc.

We would like to thank the reviewer for this suggestion, and we agree it would be very useful to use MR to explore causal effects of key risk factors. While for some of the risk factors the reviewer has listed, there are strong genetic instruments, this is not true for all. Furthermore, the UKBB has been a major contributor to recent large GWAS efforts that have supported identification of strong genetic instruments, meaning we have considerable overlap between samples, leading to overfitting/exaggeration of results. For this reason, we have now undertaken a 2-sample MR to explore the effects of BMI, smoking and alcohol use, for which there is strong evidence for being a risk factor for miscarriage from systematic reviews and meta-analyses¹²⁻¹⁴ and also strong genetic instruments^{15,16}. These analyses suggest smoking may causally increase the risk of SM, but we cannot exclude the possibility of horizontal pleiotropy explaining some of this effect. There does not seem to be strong evidence of an effect of BMI or alcohol on SM; however, these results need to be interpreted cautiously. For instance, the association between BMI and miscarriage may follow a U-shaped curve and we may not pick it up with a linear model. We have added these analyses to the manuscript main text (page 16-17).

As to the significant outcomes on Figure S7 (and Table S10), that reached Bonferroni corrected levels of statistical significance ($P < 2.93 \times 10^{-6}$), one outcome related to alcoholism and one related to post-traumatic stress disorder. However, both of these were single items from instruments that included 11 items (alcohol use questionnaire) and 21 items (post-traumatic/traumatic event questionnaire), respectively, with none of the other items reaching suggestive thresholds of statistical significance. The third outcome to show association below this p-value threshold was a job coding (scenery designer or costume designer) that is one of which lies in 42-item employment history category (MR analyses did not suggest effects on any other jobs in this list). This explanation is also present in the Methods section, but for clarity, we have added a brief comment to the respective figure legend as well.

9. Page 16, para 3. "(13% of them protein coding)" Clarify that this is a surprisingly small proportion.

As we rewrote the results section quite extensively in the light of our recent candidate gene mapping effort, we have now removed this remark from the results section.

Reviewer #2 (Remarks to the Author):

Laisk and co-authors report the first large-scale GWAS for sporadic (69,054 cases from five ancestries) and recurrent (750 cases of European ancestry) miscarriage (359,469 controls). They identify genome-wide significant associations for sporadic miscarriage (n=1, p=3.2X10e8 with OR=1.4) and recurrent miscarriage (n=3, p=1.3X10e8 with OR=1.7, p=5.2X10e9 with OR=3.4, p=2.8X10e8 with OR=3.8). The

study is a comprehensive analysis of the genetic architecture of miscarriage with Mendelian randomization, heritability and genetic correlation analyses. Despite the importance of miscarriage in women's health, there are concerns about this study. Rigorous group definitions are fundamental for validity of the findings.

Introduction

p. 5: The authors introduce sporadic miscarriage as 1-2 miscarriages and recurrent as >3 consecutive miscarriages; however, it should be noted that this is not the current definition of sporadic and recurrent miscarriage as supported by clinical guidelines (i.e., ASRM, ESHRE). Further in the Introduction (p. 6), the authors state that current guidelines define recurrent miscarriage as the loss of >2 pregnancies, which is correct. Also, on p. 6 they state that in the present study they define recurrent miscarriage as having had >3 self-reported miscarriages or code N96 (i.e., >2 miscarriages) such that they will capture the more severe end of the phenotypic distribution to differentiate it from sporadic miscarriage. This is confusing for the reader.

It is true that according to current ESHRE and ASRM guidelines, recurrent miscarriage/N96 is defined as 2 or more spontaneous abortions. However, this has not always been the case, and the most recent set of ESHRE guidelines was published in November 2017. Before that, the recurrent pregnancy loss definition varied from two miscarriages (ASRM Practice Committee, 2013⁵; Zegers-Hochschild et al., 2009⁶), to three or more consecutive pregnancy losses (Jauniaux et al., 2006⁷; RCOG Green Top Guideline, 2011⁸). All the recurrent miscarriage cohorts (UKBB, EGCUT, ALPSAC) included in this study are from Europe and have received the N96 diagnosis before 2018, therefore we believe it is acceptable to combine the N96 with self-reported data.

We have rephrased the introduction on page 6 to reflect this and be less confusing.

p. 6: The authors state that miscarriage increases with maternal age, which is well known to be associated with increased aneuploidy. Is there any consideration of the U-shaped distribution in maternal age for aneuploidy? Is that relevant in the different populations used in the study? Is there any exclusion, correction or stratification for maternal age? This is fundamental for the analysis.

It would indeed be interesting to carry out an association analysis for miscarriage stratified by maternal age at the event. However, it is difficult to do in a biobank setting when data have been collected retrospectively, and no date is recorded for self-reported events. Unfortunately, the size of the cohorts where we could access the dates for miscarriage-associated ICD codes is still insufficient for a well-powered genetic association study, but hopefully this will be possible in the future. However, the woman's year of birth has been used as a covariate in GWAS to account for potential differences associated with different time-dependent trends in reproductive behavior.

Results and Discussion

p. 7 Please replace "...on chromosome 7p15.2..." with "...on chromosome 7 in band p15.2...". There is no chromosome 7p15.2.

We thank the reviewer for finding and highlighting this, we have made the change.

p. 18 Please replace "...ZDHC13, MRGPRX2..." with "...ZDHC13 and MRGPRX2...".

We thank the reviewer for finding and highlighting this, we have made the change.

p. 18 Please replace “...HSF2BP, RRP1B...” with “...HSF2BP and RRP1B...”.

We thank the reviewer for finding and highlighting this, we have made the change.

Methods

p. 20 Is the question posed in the UKBB, “Have you ever had any stillbirths, spontaneous miscarriages or terminations?” problematic. The etiologies of stillbirth could be very different from those resulting in miscarriages, and social abortions (i.e., terminations) could be frequent and not have any underlying genetic risk. This grouping could be very heterogenous and affect the quality of the results. Were social abortions excluded?

We apologize for this confusion. In the UKBB, the self-reported cases were identified using the questionnaire field 3839 “How many spontaneous miscarriages?”, which is collected from participants who indicated they had had a stillbirth, spontaneous miscarriage or termination, as defined by their answers to data field 2774 “Have you ever had any stillbirths, spontaneous miscarriages or terminations?”. While some of the miscarriage cases might also have additional stillbirths or social abortions, the miscarriage case status was determined solely based on their response to questionnaire field 3839. We have now edited to manuscript main text to make this clearer (page 22).

p. 20 Please replace “...participants, data has...” with “...participants, data have...”.

We thank the reviewer for finding and highlighting this, we have made the change.

p. 23 and p. 37 There is a reference to the QIMR Endo samples but no previous mention of an analysis of endometriosis. The authors describe MR for endometriosis and recurrent miscarriage (p. 37) but this is given in the Methods and is not present in the Results and Discussion.

The QIMR Endo represents an Australian cohort of women with a confirmed surgical diagnosis of endometriosis, and for whom detailed reproductive history data are available. We did not conduct a separate analysis for endometriosis; however, it is known that endometriosis is associated with an increased risk of miscarriage¹⁷. For this reason, we were cautious at first and performed the meta-analysis in parallel with and without QIMR Endo cohort, respectively. However, the genome-wide significant signal for sporadic miscarriage on chromosome 13 (Figure 2) shows no significant heterogeneity in terms of effect estimate, and the results from QIMR Endo are in line with those from other cohorts, therefore we decided to keep the QIMR Endo cohort in the analysis.

The two-sample MR for endometriosis was undertaken because the top 10 lowest P-values from the PheWAS analysis included a suggestive causal effect of recurrent miscarriage on endometriosis of the uterus ($P=5.9 \times 10^{-5}$), which was not confirmed in the two-sample MR analysis. Please note that the QIMR Endo samples were not included in the recurrent miscarriage analysis.

p. 26 Please replace “...sequencing data is...” with “...sequencing data are...”.

We thank the reviewer for finding and highlighting this, we have made the change.

p. 27 The case definitions continue to be confusing from that stated in the Introduction. The authors define sporadic miscarriage here by two standards (e.g., self-report or two other ICD-10 codes), which potentially results in combining patients with different etiologies. The authors should clearly state the number in each

category. Recurrent miscarriage is defined by four different categories including code N96, which can be defined by only 2 miscarriages. Also, what is the relevance in the recurrent classification of requiring miscarriages to be consecutive? If there were an underlying genetic etiology that resulted in an increased risk for chromosomal non-disjunction (e.g., spindle fiber disorder) and there were multiple aneuploid miscarriages (e.g., trisomy 21, trisomy 18, and trisomy 13 as has been observed in single patients) separated by euploid pregnancies, would this not qualify as recurrent miscarriage? What would the results of the study be had sporadic miscarriage been defined as a single miscarriage and recurrent miscarriage as the loss of >2 pregnancies? This is critical and the results are impacted by mixing categories of patients.

Our GWAS meta-analysis for sporadic miscarriage included cohorts, where cases were identified either using self-reported data, electronic health records, or a combination of both (Supplementary Table 1, depending on what type of data was available). As a result, out of the 69,054 sporadic miscarriage cases, 26,044 were identified using only self-reported data; 1,231 only electronic health records; and 69,054 using both (in this category, majority have self-reported sporadic miscarriage). This information has now also been added to the Material & Methods section.

For recurrent miscarriage, we had data from three cohorts. ALSPAC cases (n=219) were identified using only self-reported data, whereas in UKBB and EGCUT, cases were identified using a combination of self-reported data and electronic health records. In the UKBB, which is the largest contributing cohort to the recurrent miscarriage analysis, we identified a total of 458 recurrent miscarriage cases (all ethnicities, including 421 White British cases, who were included in the final GWAS). Of these, 21 had the N96 diagnosis code, while 150 had reported at least five miscarriages, one live birth, no elective terminations (to ensure the consecutiveness of miscarriages) and 291 had reported at least 3 miscarriages, no live births or elective terminations. In the Estonian Biobank (EGCUT) cohort, we had 113 recurrent miscarriage cases. Due to the high prevalence of elective abortions in this cohort, nearly all cases were identified using the N96 diagnosis code.

It is worth noting that for all of the reported associated loci, we don't see significant heterogeneity across cohorts, i.e. depending on the data used for phenotype definition. Therefore, we believe that at least for the reported associations, the results are not impacted by combining self-reported and electronic health record-derived cases.

As noted above, the definition of recurrent miscarriage has changed over time. Since the previous diagnostic guidelines, which were in use in Europe at the time the current study was designed (and when majority of the patients had received their diagnosis) defined recurrent miscarriage as three or more consecutive pregnancy losses (Jauniaux et al., 2006, PMID: 16707507; RCOG Green Top Guideline, 2011), this was also the criteria we used. Of note, the ESHRE 2018 guidelines acknowledged that using two miscarriage as the definition of recurrent miscarriage was at least in part to facilitate research, improve shared decision making with patients and provide them psychological support, rather than because of more specific evidence that this defined a unique phenotype. In fact, the ESHRE guidelines acknowledge not all guideline group members agreed with this definition demonstrating the continuing uncertainty in the field regarding the definition of RM. Data support increasing numbers of previous miscarriages as being a risk factor for further miscarriage⁹ and increasing likelihood of aneuploid miscarriage¹⁰. Thus our rationale behind using a definition of at least 3 consecutive miscarriages (which was also the definition of recurrent miscarriage in much of Europe at the time the study was started) was to capture the more severe end of the phenotypic spectrum, which would allow to better assess whether recurrent and sporadic events lie on the same phenotypic spectrum.

To further assess which would be the most appropriate cut-off for defining different miscarriage phenotypes, we used the UKBB data (which is the largest contributing cohort) to evaluate the effect of having a different number of miscarriages on overall fertility (in this context measured a) number of pregnancies and b) as the number of children normalized by the number of pregnancies, i.e. number of children/number of pregnancies). From Figure 1 (see below) it can be seen that if we look at the number of miscarriages without considering whether they are consecutive or not, higher number of pregnancies also corresponds to a higher number of miscarriages.

A

B

Figure 1. A) Distribution of number of pregnancies according to miscarriage group; B) Distribution of number of children (normalized by the number of pregnancies) according to miscarriage group. Blue and red dots correspond to median and mean of the values, respectively.

Given that the phenotypes “1 miscarriage” and “2 miscarriages” seem to be quite similar in terms of fertility, and the group we used for recurrent miscarriage (“3 or more consecutive miscarriages”) differs from the others, we argue that redefining the recurrent miscarriage as 2 or more miscarriages and re-running the genetic analyses based on that is not justified, as our aim was to capture the more severe end of the phenotypic spectrum, which would allow to better assess whether recurrent and sporadic events lie on the same phenotypic spectrum.

We agree that the debate around how to best define recurrent miscarriage is still ongoing and perhaps additional genetic studies or studies using the rich phenotype data available in biobanks can also shed some light into this issue.

p. 28 Is there a bullet point missing before the second indented criterion “women with any of diagnoses for...” under Exclusion criteria.

Thanks for noticing and the missing bullet point has been added.

p. 36 Please replace “...one of a which lies in 42-item...” with “...one of which lies in a 42-item...”.

We thank the reviewer for finding and highlighting this, we have made the change.

p. 36 Please replace, “The nature of PheWAS are that...” with “The nature of PheWAS is that...”.

We thank the reviewer for finding and highlighting this, we have made the change.

p. 36 Please replace “...This approach is not able to conclusively determine the...” with “...This approach is not able to determine conclusively the...”.

We thank the reviewer for finding and highlighting this, we have made the change.

p. 39 The authors describe that they use Hi-C in ovaries but no description is provided of the specific cell line nor from where the data have been obtained. They also refer that for Fig. S9 they use endothelial progenitors without describing the cell line or tissue used. Are the TADs different from standard tissues (e.g., LCLs or fibroblasts)?

All Hi-C maps have been generated using the 3D Genome Browser, which allows to explore over 300 publicly available chromatin interaction data of different types. The data for ovaries and endothelial progenitors were obtained from Schmitt et al 2016¹⁸ and Javierre et al 2016¹⁹ available via 3D Genome Browser, and the corresponding references have now also been added to the manuscript. In general, TADs in selected tissues are not substantially different from other tissues. It has previously been shown that across 21 primary human tissues and cell types, TADs were highly conserved in different tissues¹⁸.

p. 40 Please italicize the complete gene symbol TLE1.

We thank the reviewer for finding and highlighting this, we have made the change.

p. 41 Please insert “an” between “with imputation”.

We thank the reviewer for finding and highlighting this, we have made the change.

p. 41 Please insert “or” prior to “genotype”.

We thank the reviewer for finding and highlighting this, we have made the change.

Supplementary Text and Figures

p. 9 and 10 What is Symptomes?

We thank the reviewer for pointing out this typo - it should be “Symptoms”, referring to ICD10 codes R00-R99. We have now made respective corrections to the figures.

References

1. Karczewski, K. J. *et al.* Variation across 141,456 human exomes and genomes reveals the spectrum of loss-of-function intolerance across human protein-coding genes. *bioRxiv* 531210 (2019). doi:10.1101/531210
2. Muñoz-Fuentes, V. *et al.* The International Mouse Phenotyping Consortium (IMPC): a functional catalogue of the mammalian genome that informs conservation. *Conserv. Genet.* **19**, 995–1005 (2018).
3. Bult, C. J. *et al.* Mouse Genome Database (MGD) 2019. *Nucleic Acids Res.* **47**, D801–D806 (2019).
4. Dawes, R., Lek, M. & Cooper, S. T. Gene discovery informatics toolkit defines

- candidate genes for unexplained infertility and prenatal or infantile mortality. *NPJ genomic Med.* **4**, 8 (2019).
5. Definitions of infertility and recurrent pregnancy loss: A committee opinion. *Fertil. Steril.* **99**, 63 (2013).
 6. Zegers-Hochschild, F. *et al.* The International Committee for Monitoring Assisted Reproductive. *Hum. Reprod.* **24**, 2683–2687 (2009).
 7. Jauniaux, E., Farquharson, R. G., Christiansen, O. B. & Exalto, N. Evidence-based guidelines for the investigation and medical treatment of recurrent miscarriage. *Hum. Reprod.* **21**, 2216–22 (2006).
 8. Recurrent Miscarriage, Investigation and Treatment of Couples (Green-top Guideline No. 17). Available at: <https://www.rcog.org.uk/en/guidelines-research-services/guidelines/gtg17/>. (Accessed: 2nd January 2020)
 9. Magnus, M. C., Wilcox, A. J., Morken, N. H., Weinberg, C. R. & Håberg, S. E. Role of maternal age and pregnancy history in risk of miscarriage: Prospective register based study. *BMJ* **364**, (2019).
 10. Ogasawara, M., Aoki, K., Okada, S. & Suzumori, K. Embryonic karyotype of abortuses in relation to the number of previous miscarriages. *Fertil. Steril.* **73**, 300–304 (2000).
 11. Leitsalu, L. *et al.* Cohort Profile: Estonian Biobank of the Estonian Genome Center, University of Tartu. *Int. J. Epidemiol.* **44**, 1137–1147 (2015).
 12. Aune, D., Saugstad, O. D., Henriksen, T. & Tonstad, S. Maternal body mass index and the risk of fetal death, stillbirth, and infant death: a systematic review and meta-analysis. *JAMA* **311**, 1536–46 (2014).
 13. Sundermann, A. C. *et al.* Alcohol Use in Pregnancy and Miscarriage: A Systematic Review and Meta-Analysis. *Alcohol. Clin. Exp. Res.* **43**, 1606–1616 (2019).
 14. Pineles, B. L., Park, E. & Samet, J. M. Systematic review and meta-analysis of miscarriage and maternal exposure to tobacco smoke during pregnancy. *Am. J. Epidemiol.* **179**, 807–23 (2014).
 15. Locke, A. E. *et al.* Genetic studies of body mass index yield new insights for obesity biology. *Nature* **518**, 197–206 (2015).
 16. Liu, M. *et al.* Association studies of up to 1.2 million individuals yield new insights into the genetic etiology of tobacco and alcohol use. *Nat. Genet.* **51**, 237–244 (2019).
 17. Farland, L. V. *et al.* Endometriosis and Risk of Adverse Pregnancy Outcomes. *Obstet. Gynecol.* **134**, 527–536 (2019).
 18. Schmitt, A. *et al.* A Compendium of Chromatin Contact Maps Reveals Spatially Active Regions in the Human Genome. *Cell Rep.* **17**, 2042–2059 (2016).
 19. Javierre, B. M. *et al.* Lineage-Specific Genome Architecture Links Enhancers and Non-coding Disease Variants to Target Gene Promoters. *Cell* **167**, 1369-1384.e19 (2016).

Additional Figures

Examples of chromatin interactions in different tissues for the sporadic miscarriage (rs146350366) and recurrent miscarriage signals (rs7859844).

rs146350366

Lung (Ren lab data)

Hippocampus (Ren lab data)

Liver (Ren lab data)

Aorta (Ren lab data)

Left Ventricle (Ren lab data)

rs7859844

MSC (Ren lab data)

Ovary (Ren lab data)

Aorta (Ren lab data)

Bladder (Ren lab data)

Adrenal Gland (Ren lab data)

CD34 (data from Luscombe & Osborne)

GM12878 (data from Luscombe & Osborne)

H1ESC (Ren lab data)

IMR90 (Ren lab data)

Liver (Ren lab data)

Lung (Ren lab data)

Reviewers' comments:

Reviewer #1 (Remarks to the Author):

1. The authors argue that current definitions of 'recurrent miscarriage' are somewhat arbitrary and their aim was to "assess whether recurrent and sporadic events lie on the same phenotypic spectrum" which I agree is an important aim. But unfortunately their binary categorisation of cases into (relatively very severe) recurrent and sporadic events means they have too few recurrent miscarriage cases to address this key aim. From the numbers they state, of 273,472 UKBB women, 11% had 1 event, 2.8% had 2 events, and 0.2% had 'recurrent' events. Describe the association between the 4 identified signals on the risk of 1, 2, 3+ events separately at least in UK Biobank. Also describe the SNP based heritability of 1 and 2 events separately.

We have now carried out separate GWAS analyses in the UK Biobank for women with 1, 2 or 3+ miscarriages, respectively and summarised the findings in the table below. Marker rs146350366, identified in the sporadic miscarriage analysis, was nominally significant in both the "1 miscarriage" (P= 9.2×10⁻⁵) and "2 miscarriages" (P=0.017) analyses, but was statistically not significant (and had opposite effect direction) in the 3 or more miscarriage analyses. Two (rs7859844, rs183453668) of the three signals from multiple consecutive miscarriage analysis were nominally significant also in the "3 or more miscarriages" analyses (P=0.01 and P=0.04, respectively), but showed no evidence of association with the "1 miscarriage" and "2 miscarriages" phenotypes. Collectively, this indicates that for the reported associations, at least in the UKBB dataset, the "2 miscarriages" phenotype is more similar in terms of effect estimate to the "1 miscarriage" group than the subset of women who have three or more miscarriages and justifies the joint analysis of women with 1-2 miscarriages.

We also used the GWAS summary statistics to calculate the heritability separately for single miscarriage (h²= 0.0043 +/-0.0059) and 2 miscarriages (0.0353 +/-0.019)) on the liability scale, assuming a population prevalence of 20%.

These results have now also been added to the Supplementary Text of the manuscript as a part of a more thorough explanation on used phenotype definitions.

Table 1. Association between the reported GWAS signals and different miscarriage phenotypes in the UKBB

Variant	1 miscarriage (n_cases=29414)	2 miscarriages (n_cases=7238)	3+ miscarriages (n_cases=3890)	3+ consecutive miscarriages (n_cases=420)
Sporadic miscarriage (1-2 miscarriages)				
rs146350366	p=9.2e-05 OR=1.38 (1.18-1.45)	p=0.017 OR=1.45 (1.08-1.94)	p=0.087 OR=0.70 (0.46-1.05)	p=0.051 OR=0.30 (0.09 – 1.005)

Multiple consecutive miscarriage (3+ consecutive miscarriages)				
rs7859844	p=0.27 OR=1.02 (0.98- 1.06)	p=0.83 OR=0.99 (0.93- 1.06)	p=0.01 OR= 1.13 (1.03- 1.24)	p=4.8e-08 OR=2.29 (1.71- 3.08)
rs143445068	p=0.97 OR=0.99 (0.89- 1.11)	p=0.60 OR=1.06 (0.86- 1.31)	p=0.06 OR=1.31 (0.98- 1.74)	p=3.5e-05 OR=8.3 (3.1-22.6)
rs183453668	p=0.13 OR=1.11 (0.97- 1.28)	p=0.56 OR=0.92 (0.70- 1.21)	p=0.04 OR=1.48 (1.02- 2.14)	p=2.9e-04 OR1.9 (3.15-45.28)

2. Add description and comment on current definitions of sporadic and recurrent miscarriage. In the Introduction Line 110 it is not appropriate to simply state their own definitions here which gives the impression that these are established definitions.

We have added the description and discussion around the current definitions of sporadic and recurrent miscarriage. Moreover, due to the lack of consensus in the field, we have now renamed our phenotypes to “sporadic miscarriage” and “multiple consecutive miscarriage”, and rewritten the draft accordingly.

3. If phenotypic associations are to be retained, the findings need clarification. Line 250/251: How can associations support the association with more live births if models are adjusted for number of live births?

We thank the reviewer for pointing out this misleading statement, which has now been removed.

4. The MR models to assess potential determinants of SM is welcome. The same should be performed for recurrent miscarriage.

We agree with the reviewer that adding the MR models to assess potential determinants of sporadic miscarriage has improved the manuscript. However, the current sample size for multiple consecutive miscarriage is insufficient to support a well-powered MR analysis for the risk factors of recurrent miscarriage, therefore we feel a similar analysis for this phenotype would be underpowered and potentially misleading.

Reviewer #2 (Remarks to the Author):

A remaining issue is the definition of recurrent miscarriage for the analysis. Given the history of uncertainty in the definition of recurrent miscarriage over time and some heterogeneity in the cohorts, perhaps an appropriate analysis to consider would be to separate cases by numbers of miscarriage: one miscarriage (sporadic), two miscarriages and separately 3 or greater, or perhaps to consider combining one and two miscarriages in a group for comparison to a group with two and three miscarriages reasoning that the category of two miscarriages will contain both cases that are sporadic and recurrent. No doubt, this will work against a well-powered GWAS, but might reduce some of the overlap in classification.

*In a further attempt to justify the currently used phenotype definitions, we have explored and described the observed genetic associations in different miscarriage categories (1 miscarriage, 2 miscarriages, 3 or more miscarriages, 3 or more **consecutive** miscarriages) in the UKBB. As can be seen from Table 1, based on the four reported genome-wide significant signals, the group “2 miscarriages” is more similar (in terms of effect estimates) to “1 miscarriage” (see the effect estimates for rs146350366). Similarly, if we look at the three hits from the recurrent miscarriage analysis, we can see that two of these variants are nominally significant also in the “3 or more miscarriages” group, but nonsignificant in the “2 miscarriages” group. Combining this with the phenotype level analyses where we see that the “1 miscarriage” and “2 miscarriage” groups are very similar in terms of reproductive characteristics, we would prefer to keep the current division. Moreover, due to the lack of consensus in the field, we have now renamed our phenotypes to “sporadic miscarriage” and “multiple consecutive miscarriage”, and rewritten the draft accordingly.*

Reviewers' comments:

Reviewer #1 (Remarks to the Author):

1. The authors argue that current definitions of 'recurrent miscarriage' are somewhat arbitrary and their aim was to "assess whether recurrent and sporadic events lie on the same phenotypic spectrum" which I agree is an important aim. But unfortunately their binary categorisation of cases into (relatively very severe) recurrent and sporadic events means they have too few recurrent miscarriage cases to address this key aim. From the numbers they state, of 273,472 UKBB women, 11% had 1 event, 2.8% had 2 events, and 0.2% had 'recurrent' events. Describe the association between the 4 identified signals on the risk of 1, 2, 3+ events separately at least in UK Biobank. Also describe the SNP based heritability of 1 and 2 events separately.
2. Add description and comment on current definitions of sporadic and recurrent miscarriage. In the Introduction Line 110 it is not appropriate to simply state their own definitions here which gives the impression that these are established definitions.
3. If phenotypic associations are to be retained, the findings need clarification. Line 250/251: How can associations support the association with more live births if models are adjusted for number of live births?
4. The MR models to assess potential determinants of SM is welcome. The same should be performed for recurrent miscarriage.

Reviewer #2 (Remarks to the Author):

Laisk and co-authors have provided a thorough and thoughtful response to the reviewer comments and submitted a revised manuscript. Although they have tried to extract as much biological insight as possible from their data into the genetic architecture of sporadic and recurrent miscarriage, they remain stymied by the finding of only one significant signal for sporadic miscarriage and three signals for habitual miscarriage.

A collection of Promoter Capture Hi-C data from the 3D Genome Browser are provided in available tissue types for a region in high LD with tag SNP rs146350366 that is in contact with the FGF9 promoter in many of the tissues. The authors address reviewer questions around the selection of endothelial progenitors in the context of the 3D structure not being specific to a single tissue but also acknowledge that it is not a confirmation of a target tissue. A further analysis incorporating pLI and data from mouse models led to a focus on candidate genes in all four loci linked to placental biology, and overall this may be true, but it remains speculative and not well defined, lacking the evidence that one would have hoped to have found to make a compelling story.

A remaining issue is the definition of recurrent miscarriage for the analysis. Given the history of uncertainty in the definition of recurrent miscarriage over time and some heterogeneity in the cohorts, perhaps an appropriate analysis to consider would be to separate cases by numbers of miscarriage: one miscarriage (sporadic), two miscarriages and separately 3 or greater, or perhaps to consider combining one and two miscarriages in a group for comparison to a group with two and three miscarriages reasoning that the category of two miscarriages will contain both cases that are sporadic and recurrent. No doubt, this will work against a well-powered GWAS, but might reduce some of the overlap in classification.

In sum, the manuscript does not provide much further insight into the genetic risk for sporadic vs recurrent miscarriage and lacks also in providing data with sufficient evidence to carry forward biological investigations based on the limited number of associations.

REVIEWERS' COMMENTS:

Reviewer #1 (Remarks to the Author):

I thank the authors for addressing the question of case definitions with additional analyses which shed some helpful insights and support their proposed categorisation. However, these have been added to the Supplement with no mention at all in the main text (apologies if I missed it) except surprisingly in the final concluding paragraph they highlight that the genetic basis for the different definitions "does not seem to overlap". They should add at least a brief line to the main text results to say that the locus for sporadic cases was completely null for multiple recurrent cases, and vice versa.

Also in support of the hypothesis that these categories have a different genetic and biological basis, they might wish to highlight the contrasting findings that sporadic cases are associated with more live births, but multiple recurrent cases are associated with infertility.

Page 7 - They define "multiple consecutive miscarriage as having had ≥ 3 self-reported miscarriages". But I presume here they must mean ' ≥ 3 self-reported recurrent miscarriages'. Getting these definitions right is critical.

Page 15 (para 2) Insert 'phenotypic' to indicate this is a change of approach. "We also examined phenotypic associations..."

Reviewers' comments:

Reviewer #1 (Remarks to the Author):

I thank the authors for addressing the question of case definitions with additional analyses which shed some helpful insights and support their proposed categorisation. However, these have been added to the Supplement with no mention at all in the main text (apologies if I missed it) except surprisingly in the final concluding paragraph they highlight that the genetic basis for the different definitions "does not seem to overlap". They should add at least a brief line to the main text results to say that the locus for sporadic cases was completely null for multiple recurrent cases, and vice versa.

Also in support of the hypothesis that these categories have a different genetic and biological basis, they might wish to highlight the contrasting findings that sporadic cases are associated with more live births, but multiple recurrent cases are associated with infertility.

We thank the reviewer for their concern. In the final version of the manuscript, we have added references to the additional analyses in both the Introduction and Methods sections. Also, we have added the paragraph „To clarify the potential genetic overlap between miscarriage phenotypes, we performed a cross-phenotype look-up of the associated loci. All the multiple consecutive miscarriage loci were statistically not significant in the spontaneous miscarriage analysis, and vice versa (see Supplementary Note 1). This indicates the genetic basis of sporadic and multiple consecutive miscarriage does not overlap, at least not for the reported loci, and lends further support to our phenotype definitions“ to the Results section.

We have also added additional sentences to the Results section (namely the 'Miscarriage genetically correlated with number of children' and 'Miscarriage is associated with a variety of health outcomes' subsections) and Discussion to highlight the contrast in the used phenotypes.

Page 7 - They define "multiple consecutive miscarriage as having had ≥ 3 self-reported miscarriages". But I presume here they must mean ' ≥ 3 self-reported recurrent miscarriages'. Getting these definitions right is critical.

We assume the reviewer means ' ≥ 3 self-reported consecutive miscarriages' instead and have now added this specification to page 7.

Page 15 (para 2) Insert 'phenotypic' to indicate this is a change of approach. "We also examined phenotypic associations..."

We have now added 'phenotypic' to the sentence.